# Heterozygous missense variant in *GLI2* impairs human endocrine pancreas development

Laura M. Mueller[1], Abigail Isaacson[1], Heather Wilson[1], Anna Salowka[1], Isabel Tay[1], Maolian Gong[2,3], Nancy Samir Elbarbary[4], Klemens Raile[2,3] & Francesca M. Spagnoli ●[1] ✉

Missense variants are the most common type of coding genetic variants. Their functional assessment is fundamental for defining any implication in human diseases and may also uncover genes that are essential for human organ development. Here, we apply CRISPR-Cas9 gene editing on human iPSCs to study a heterozygous missense variant in *GLI2* identified in two siblings with early-onset and insulin-dependent diabetes of unknown cause. GLI2 is a primary mediator of the Hedgehog pathway, which regulates pancreatic β-cell development in mice. However, neither mutations in *GLI2* nor Hedgehog dysregulation have been reported as cause or predisposition to diabetes. We establish and study a set of isogenic iPSC lines harbouring the missense variant for their ability to differentiate into pancreatic β-like cells. Interestingly, iPSCs carrying the missense variant show altered GLI2 transcriptional activity and impaired differentiation of pancreatic progenitors into endocrine cells. RNA-Seq and network analyses unveil a crosstalk between Hedgehog and WNT pathways, with the dysregulation of non-canonical WNT signaling in pancreatic progenitors carrying the GLI2 missense variant. Collectively, our findings underscore an essential role for GLI2 in human endocrine development and identify a gene variant that may lead to diabetes.

Diabetes is a worldwide health problem caused by the loss or dysfunction of the insulin-secreting β-cells in the pancreas[1,2]. Unelucidated forms of monogenic diabetes, arising from rare mutations in one single gene, represent invaluable models for identifying new targets of β-cell development and function[3]. Notably, monogenic diabetes predominantly results from mutations in transcription factors, which are involved in pancreas development[4]. Moreover, genetic variants in or nearby some of these transcription factor genes have been associated with type 2 diabetes risk[5].

To date, over 30 genes have been linked to monogenic diabetes, yet many patients are misdiagnosed. Additionally, the disease penetrance can vary substantially among individuals with the same mutation[6]. Recent studies have underscored the benefits of genetic diagnosis in patients with monogenic diabetes, enabling to improve patient care by optimizing treatment strategy, to predict the disease course and to define risks for relatives[3,6]. Therefore, identification of novel disease-related loci might instruct translational efforts towards new treatment of diabetes and help in defining genetic factors that increase susceptibility to type 2 diabetes.

[1]Centre for Gene Therapy and Regenerative Medicine, King's College London, Great Maze Pond, London SE1 9RT, United Kingdom. [2]Department of Pediatric Endocrinology and Diabetology, Charité, Berlin, Germany. [3]Experimental and Clinical Research Center (ECRC), Charité Medical Faculty, Max-Delbrueck-Center for Molecular Medicine (MDC), Berlin, Germany. [4]Department of Pediatrics, Diabetes and Endocrine Unit, Faculty of Medicine, Ain Shams University, Cairo, Egypt. ✉e-mail: francesca.spagnoli@kcl.ac.uk

Human pluripotent stem cell (hPSC) models are increasingly being used to study monogenic diabetes, as rodent models have often failed to recapitulate disease phenotypes observed in humans[4,7]. Noteworthy examples are mutations in *GATA6*, *HNF1β* (MODY5) and *HNF1α* (MODY3) genes[8–11]. For all of them, the human disease phenotype is not mirrored in mouse models, while hPSCs technology in combination with in vitro pancreatic differentiation protocols have contributed to shed light on them[7–11]. In particular, the generation of isogenic hPSC pairs, which differ only by the mutation of interest, has been key to minimize confounding results due to differences in genetic backgrounds and ensures that observed phenotypes are attributable to a specific genetic defect[12,13].

GLI2 is a member of the glioma-associated oncogene homolog (GLI) family of transcription factors and mediates the effects of the Hedgehog (HH) pathway[14–16]. During pancreas organogenesis, repression of HH signaling is required for pancreas fate specification and cell differentiation[17], while at later stages active HH signaling is essential for the maintenance of endocrine function[18]. However, high HH levels in insulin-producing β-cells have been reported to impair their function by interfering with the mature β-cell differentiation state[19]. Thus, a precise spatiotemporal HH regulation appears to be critical for mouse pancreas development and β-cell function, whereas its function in human embryonic pancreas is not known. A number of rare heterozygous mutations in GLI2 have been associated with a spectrum of clinical phenotypes, including holoprosencephaly, hypopituitarism and other developmental disorders[20–22]. Aberrant activation of the HH pathway and elevated expression of *GLI2* has also been reported in pancreatic cancer[23,24]. Here, we applied CRISPR-Cas9 gene editing on human iPSCs to study a heterozygous missense variant of *GLI2* gene (GLI2[P1554L]) identified in a child with diabetes of unknown origin and family members. A set of isogenic iPSC lines harboring the mutation were studied for their ability to differentiate into pancreatic β-like cells in comparison to control lines. Our findings show that GLI2 plays an essential role in human endocrine cell development and identify a previously unknown variant that eventually leads to diabetes.

## Results

### Family with a history of diabetes carries a heterozygous missense variant in *GLI2*

In a consanguineous family with diabetes of unknown etiology, we identified an uncharacterized heterozygous *GLI2* missense variant (hg38 chr2:120990575; NM_001371271.1:c.4661 C > T; p.P1554L; rs767802807) (Fig. 1a, b; Supplementary Table 1). The index child developed diabetes at the age of 5 years, the individual currently in adult age requires multiple daily insulin injections and already had complications onset. The younger sibling also carries the variant and had onset of insulin-dependent diabetes at age of 3 years. Although both parents harbor the c.4661T variant in *GLI2*, only the father developed diabetes. Moreover, the grandmother from the paternal side also had diabetes. However, we were not able to determine her genotype as she was deceased. Finally, no mutation in known, diabetes-causing genes have been found in a heterozygous or homozygous manner in any of the family members.

So far, mutations in *GLI2* have not been associated with diabetes in publicly available databases. To assess the effect of the identified GLI2 c.4661 C > T variant, we applied various independent *in-silico* tools (Supplementary Table 1). The GLI2 missense variant results into a single amino acid change (p.P1554L) in the transcription activation domain (TAD) of the protein and the affected residue is highly conserved across species with a genomic evolutionary rate profiling score >3 (Fig. 1a, c). Moreover, the p.P1554L missense mutation showed high CADD score and was predicted to be deleterious by all applied bioinformatics tools[25] (Supplementary Table 1). Mutations that are pathogenic for early-onset diabetes are expected to be very rare in the

population. Accordingly, the allele frequency (AF) of p.P1554L was 0.000044 in gnomAD and MAF < 0.01[25].

Next, we examined the spatio-temporal pattern of expression of *GLI2* in human pancreas (Fig. 1e and Supplementary Fig. 1l). We found that GLI2 is present in human early fetal pancreas, specifically in a subset of PDX1+ pancreatic progenitor cells at Carnegie Stage (CS) 20 and colocalizes with PDX1 and NKX6.1 at later stages (Fig. 1e). Consistently, *GLI2* was expressed in human iPSCs undergoing differentiation into pancreatic cells, especially abundant at pancreatic progenitor (PP) and β-like cell stages (Supplementary Fig. 1l). Together, our results suggest that GLI2 is part of a complex regulatory network regulating human β-cell development.

### The GLI2 p.P1554L patient variant exhibits decreased HH signaling activation

All three GLI proteins (GLI1, GLI2 and GLI3) share a conserved C2H2-type zinc finger DNA-binding domain[15]. GLI1 functions exclusively as transcriptional activator, while GLI2 and GLI3 can act as both activators or repressors, depending on the levels of HH[16]. To start investigating the newly identified GLI2 p.P1554L (hereinafter referred to as GLI2[P>L]) patient variant, we first tested whether it interferes with the basic transactivation property of the GLI2 transcription factor using a GLI-responsive luciferase reporter assay[26]. The GLI2[P>L] variant displayed significantly reduced transcriptional activity compared to wild-type (WT) GLI2 activity (Fig. 1d).

Next, to study the GLI2[P>L] patient variant in pancreatic β-cells, we established human isogenic iPSC lines with either the heterozygous or homozygous variant and differentiated them along the pancreatic cell lineage (Supplementary Fig. 1a). We used an iPSC line, which carries a doxycycline (DOX)-inducible Cas9 expression cassette inserted into the AAVS1 locus, allowing the de novo introduction of mutations into otherwise healthy iPSCs[27]. This strategy not only bypasses the limitations of iPSC generation from patients that requires biopsy and reprogramming, but also overcomes the issues related to the genetic background difference of patients derived-iPSCs *versus* healthy controls, which can substantially influence the phenotype[12]. Both heterozygous (GLI2[P>L HET]) and homozygous (GLI2[P>L HOMO]) gene-edited iPSCs displayed normal morphology and expressed pluripotency markers (Supplementary Fig. 1b–e). Moreover, GLI2[P>L HET] iPSCs displayed reduced endogenous transcriptional activation of the GLI-responsive luciferase reporter in comparison to control iPSCs (referred to as GLI2[CTRL]) (Supplementary Fig. 2a). These results are in line with the luciferase assay in HEK293T cells (Fig. 1d), indicating that GLI2[P>L] variant impairs HH signaling.

### Defective pancreatic progenitor and endocrine fate specification in iPSCs carrying the GLI2[P>L] patient variant

The patient-like heterozygous GLI2[P>L HET] iPSCs were then differentiated into pancreatic β-like cells using a suspension-based differentiation protocol, as previously reported (Fig. 2a)[28,29]. Of note, this differentiation protocol[29] does not require the addition of any HH signaling pathway inhibitors, unlike other protocols[7,30–32].

Healthy GLI2[CTRL] iPSCs robustly differentiated along the pancreatic cell lineage, yielding >20% of insulin-secreting β-like cells at day (D) 21 of differentiation (Supplementary Fig. 1f–k). Patient-like GLI2[P>LHET] iPSCs differentiated successfully to definitive endoderm and gut tube stage, as shown by cell morphology and expression of stage-specific markers (Fig. 2b, c), with >90% of the cells PDX1-positive, as quantified by flow cytometry analysis (Fig. 2b). Patient-like GLI2[P>L HET] iPSCs also progressed to the pancreatic and endocrine progenitor stages as the GLI2[CTRL] cells (Fig. 2c, d). However, the level of expression of *PDX1* and *NKX6.1* was significantly lower in GLI2[P>LHET]-derived progenitors (Fig. 2c) and this was accompanied by ~70% reduction in the fraction of PDX1/NKX6.1-double positive cells as compared to control cells (Fig. 2d). Additionally, the

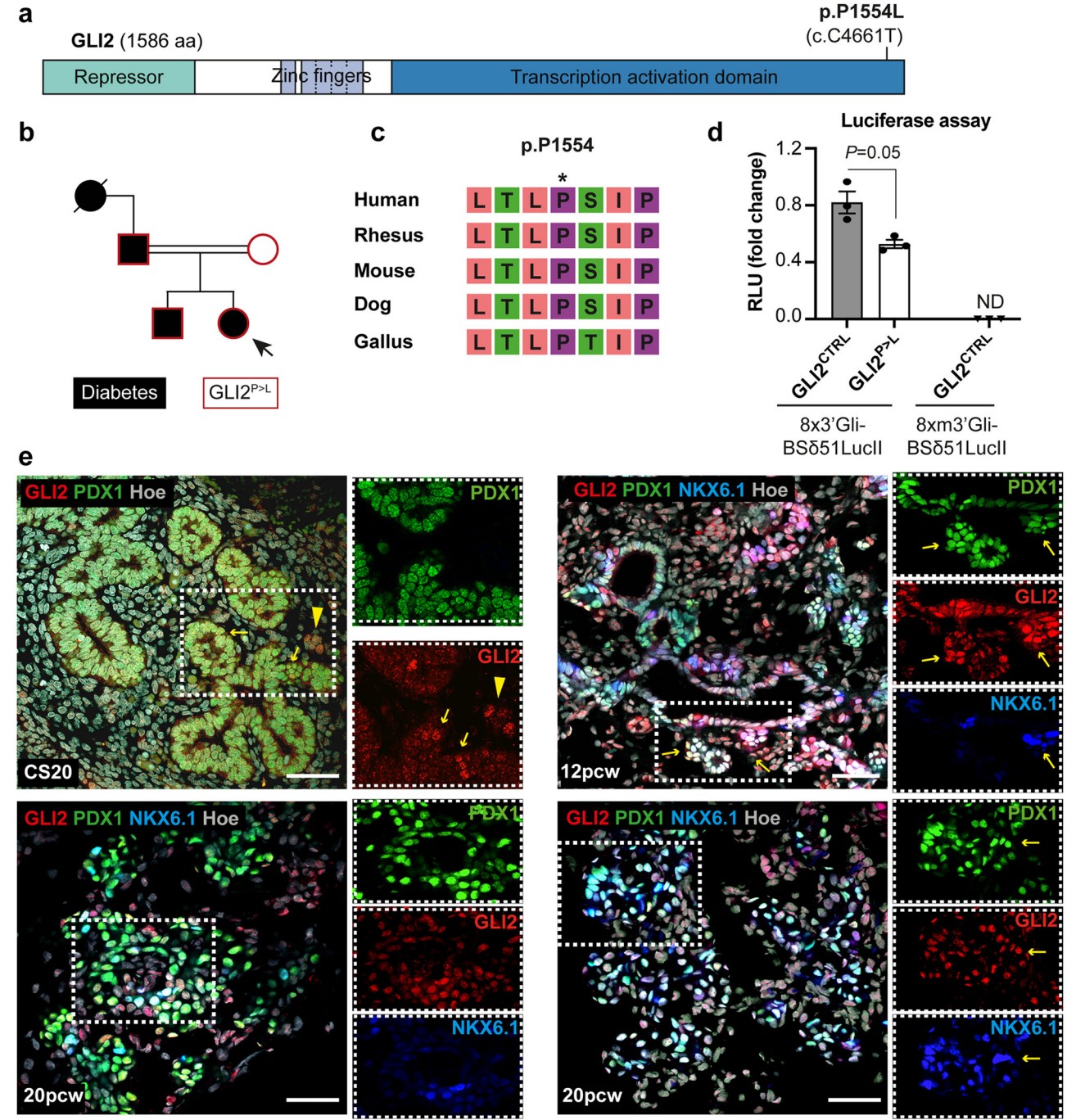

**Fig. 1 | Identification of heterozygous GLI2 mutation in a family with diabetes of unknown etiology. a** Schematic representation of human GLI2 protein and the position of the single amino acid change p.P1554L in the Transcription Activation Domain (TAD). **b** Family tree of patients with puberty-onset diabetes. The heterozygous *GLI2* p.P1554L (c.C4661T) variant was found in four individuals (red outline) of a consanguineous family with incomplete penetrance; diabetes is indicated by black shading. Females are represented by circles and males by squares. Consanguinity is represented by a horizontal parallel double bar. The missense variant was identified in patient 4 (index; arrow) by next generation sequencing[55]. The other family members were tested by Sanger sequencing. **c** The amino acid residue P1554 is highly conserved among species according to the University of California Santa Cruz (UCSC) Genome Browser. **d** Luciferase-based reporter assay with GLI-responsive construct in HEK 293 T cells. Wild-type (8 × 3'Gli-BSδ51LucII) or mutated (8xm3'Gli-BSδ51LucII) luciferase reporter plasmid was co-transfected with a Renilla luciferase control plasmid and indicated DNA expression vectors (GLI2^CTRL or GLI2^P>L). Results were normalized for transfection efficiency using Renilla luciferase and are represented as Firefly/Renilla activity ratio. The experiment was carried out three independent times with similar results. The average relative light units (RLU) of one representative experiment are shown. Two-tailed Student's *t*-test, *P* = 0.05. Values shown are mean ± SD. ND, not detected. Source data are provided as a Source Data file. **e** Representative immunofluorescent (IF) images with indicated antibody combinations on human pancreas at Carnegie Stage (CS) 20, 12 postconception week (pcw) and 20 pcw. At CS20 arrows indicate GLI2/PDX1-positive cells; arrowheads indicate GLI2-positive cell cluster next to PDX1-positive epithelium. NKX6.1 was detected at 12 pcw and 20 pcw. Arrows indicate a subset of pancreatic progenitors positive for GLI2, PDX1 and NKX6.1. Nuclei were labeled with Hoechst (Hoe). Scale bar, 20 μm.

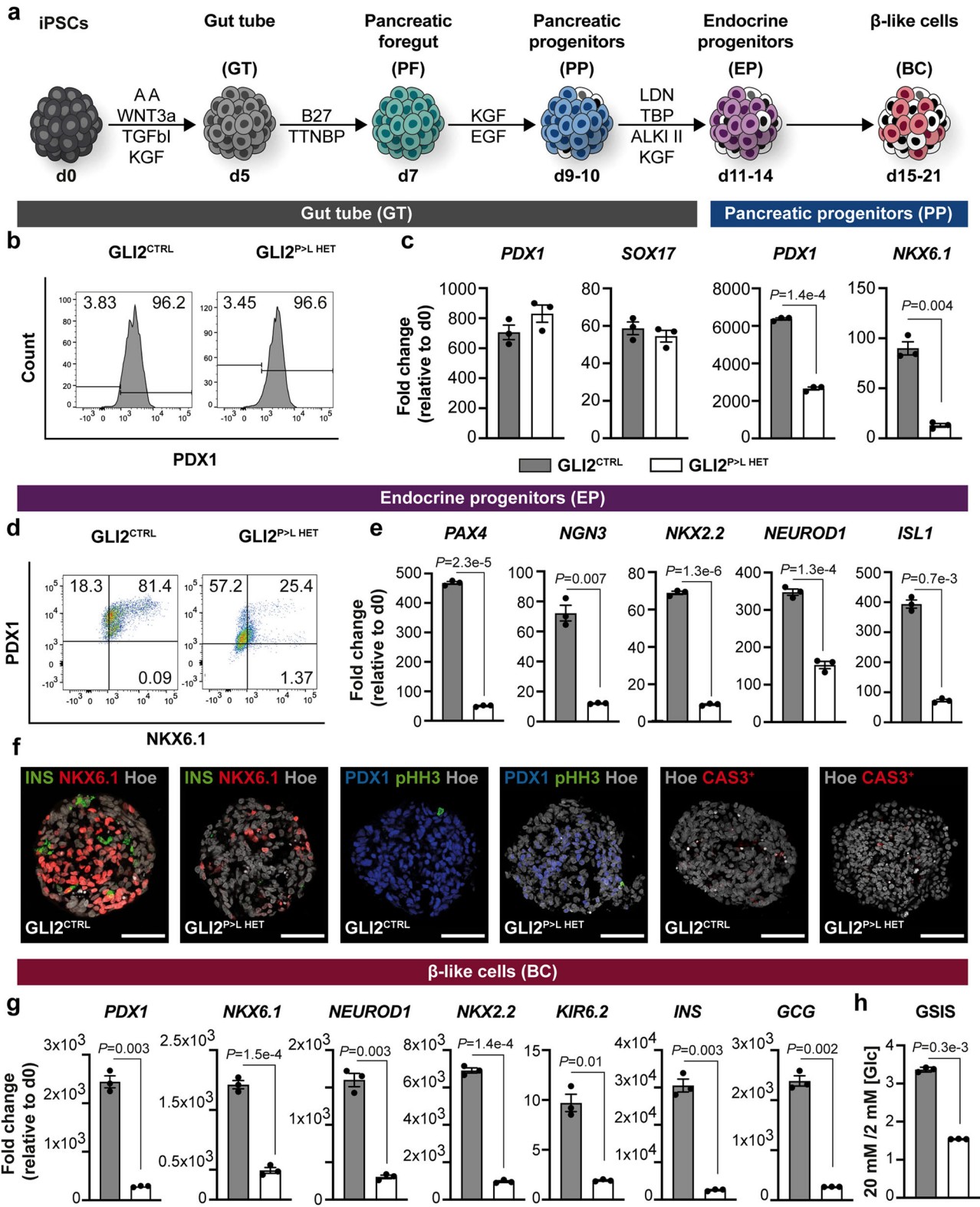

expression levels of other endocrine transcription factors, such as *PAX4*, *NGN3* and its downstream targets *NEUROD1* and *NKX2.2*, were reduced in GLI2[P>L HET]-derived endocrine progenitors (Fig. 2e). Consistently, the number of NKX6.1/INSULIN-double positive cells was significantly lower in GLI2[P>L HET]-derived cell clusters compared to controls (Fig. 2f and Supplementary Fig. 2b). No significant changes in proliferation were measured in the GLI2[P>L HET] mutant

cells compared to controls (Fig. 2f and Supplementary Fig. 2d, f), while the number of apoptotic Cleaved Caspase-3 (cCAS3)+ cells was significantly increased at D9 (Supplementary Fig. 2e, g). Altogether, our results demonstrate that the GLI2[P>L] patient variant impairs the activation of genes essential for pancreatic progenitors and subsequent endocrine development, without inducing other endodermal lineages (Supplementary Fig. 2j).

**Fig. 2 | Characterization of GLI2^{P>L HET}-derived iPSCs undergoing pancreatic endocrine differentiation. a** Schematic representation of the differentiation protocol of iPSCs into pancreatic β-like cells[29]. Cells were differentiated in suspension as 3D clusters. **b** Representative flow cytometry plots of PDX1+ cells (shown as %) in WT GLI2^{CTRL} and GLI2^{P>L HET}-derived cells at day (D) 5 of differentiation. The differentiation experiment was carried out at least three independent times. **c** RT-qPCR analysis of selected gene transcripts in GLI2^{CTRL} and GLI2^{P>L HET} differentiated cells at D5 and D9. Values are normalized to *GAPDH*. Data are shown as fold change relative to undifferentiated cells (D0). Values shown are mean ± SEM. *n* = 3 differentiation experiments on two GLI2^{P>L HET} independent clones. Two-tailed Student's *t*-test; exact *P*-values are reported in the figure. **d** Representative flow cytometry plots of NKX6.1+ and PDX1+ cells (shown as %) in GLI2^{CTRL}- and GLI2^{P>L HET}-derived endocrine progenitor (EP) cell clusters at D14. **e** RT-qPCR analysis of selected gene transcripts in GLI2^{CTRL} and GLI2^{P>L HET} differentiated cells at D14. Data are represented as fold change relative to undifferentiated cells (d0). Values shown are mean ± SEM. *n* = 3 differentiation experiments on two GLI2^{P>L HET} independent clones. Two-tailed Student's *t*-test; exact *P*-values are reported in the figure. **f** Representative IF images for indicated markers on GLI2^{CTRL}- and GLI2^{P>L HET}-derived endocrine progenitor cells. Nuclei were labeled with Hoechst (Hoe). Scale bar, 20 μm. **g** RT-qPCR analysis of selected gene transcripts in differentiated cells at D21. Data are represented as fold change relative to undifferentiated cells (D0). Values shown are mean ± SEM. *n* = 3 differentiation experiments on two GLI2^{P>L HET} independent clones. Two-tailed Student's *t*-test; exact *P*-values are reported in the figure. **h** Representative static glucose stimulation insulin secretion (GSIS) assay of D21 GLI2^{CTRL} and GLI2^{P>L HET} derived β-like cells. On *Y*-axis ratio of insulin secreted at high *versus* low glucose [Glc] conditions. Values shown are mean ± SD. *n* = 3 differentiation experiments on two GLI2^{P>L HET} independent clones. Two-tailed Student's *t*-test; exact *P*-values are reported in the figure. Source data are provided as a Source Data file.

## Dose-dependent effect of GLI2^{P>L} variant in β-cell differentiation

Co-expression of *PDX1* and *NKX6.1* in endocrine progenitors is a key step necessary for the transient expression of *NGN3* and subsequent generation of glucose-responsive β-like cells[33]. Since GLI2^{P>L} patient variant affected the differentiation of iPSCs into pancreatic and endocrine progenitors, we asked whether this had any functional consequences on β-cell development. Heterozygous GLI2^{P>L}-derived endocrine progenitor cells were cultured in suspension for an additional week to promote the formation of β-like cells and then analyzed for their differentiation state and functional properties. Consistent with the findings at earlier stages, GLI2^{P>L HET} β-like cells showed reduced expression of essential β-cell markers, including key transcription factors (*PDX1*, *NKX6.1*, *NEUROD1*, *NKX2.2*), genes important for insulin secretion (*KIR6.2*) and endocrine hormones (*INSULIN, GLUCAGON*) (Fig. 2g). Moreover, GLI2^{P>L HET} β-like cells showed a striking decrease of insulin release in response to glucose stimulation (Fig. 2h). Thus, these findings support impaired β-cell differentiation in GLI2^{P>L HET} cells, which might be responsible for causing diabetes in individuals carrying the c.C4661T variant in *GLI2*.

To further characterize the phenotype and assess a possible dose-dependent effect of the GLI2^{P>L} variant, we differentiated homozygous GLI2^{P>L} iPSC lines towards pancreatic progenitors and β-like cells. Mutant and control lines differentiated efficiently into definitive endoderm and pancreatic foregut stages (Fig. 3a, b). Quantification by flow cytometry showed that over 90% of GLI2^{CTRL} and GLI2^{P>L HOMO}-derived cells were positive for PDX1 (Fig. 3a). However, after gut tube stage, the number of GLI2^{P>L HOMO} cell clusters gradually decreased and differentiation arrested at endocrine progenitor stage, failing to reach β-like cell stage. Differences between GLI2^{CTRL} and GLI2^{P>L HOMO} became evident at pancreatic progenitor stage and exacerbated at endocrine progenitor stage, with only ~10% of GLI2^{P>L HOMO}-derived clusters being positive for both PDX1 and NKX6.1 (Fig. 3c, e). Taking together, the homozygous GLI2^{P>L} iPSCs show a more severe phenotype than GLI2^{P>L HET} iPSCs, failing to differentiate into β-like cells (Fig. 3). These findings suggest a dose-dependent effect of the GLI2^{P>L} variant on endocrine lineage differentiation. Notably, no patients homozygous for the GLI2^{P>L} variant were reported so far.

Finally, GLI2^{P>L HOMO}-derived pancreatic progenitor cells failed to progress along endocrine differentiation also when exposed to a distinct differentiation protocol which involves inhibition of HH signaling pathway (Supplementary Fig. 3)[32]. Glucagon expression was not affected in this condition (Supplementary Fig. 3c), suggesting a more specific impairment of the β-cell lineage. Thus, independently of the differentiation protocols applied, GLI2^{P>L HOMO} cultures showed decreased expression of endocrine transcription factors and β-cell genes and overall reduction in *bonafide* NKX6.1/INSULIN-positive β-cells (Figs. 2, 3 and Supplementary Fig. 3).

## Inhibition of non-canonical WNT pathway partially rescues GLI2^{P>L HET}-derived endocrine progenitors

To further characterize putative downstream mechanisms underlying the impaired endocrine differentiation of GLI2^{P>L HET} iPSCs, we performed bulk RNA-seq at three stages of differentiation (iPSC, gut tube, and endocrine progenitor stages) (Fig. 4a). At early stages (iPSC and gut tube stage), the transcriptomes of GLI2^{CTRL} and GLI2^{P>L HET}-derived cells were highly comparable, with only 48 genes significantly dysregulated at gut tube stage (Supplementary Fig. 4a; Supplementary Data 3, 4). By contrast, differences at the transcriptome level became evident at endocrine progenitor stage [928 genes upregulated and 1070 downregulated in mutant *versus* control cells] (Fig. 4b; Supplementary Data 5). Gene ontology (GO) term analysis identified genes classified under 'endocrine pancreas development', 'regulation of insulin secretion' and 'glucose homeostasis' categories to be among the most significantly downregulated ones in GLI2^{P>L HET} endocrine progenitor cells (Supplementary Fig. 4c). Consistently, gene set enrichment analysis (GSEA) showed that gene sets significantly downregulated in mutant GLI2^{P>L HET} cells belong to the 'Pancreas development' GO term, including *PDX1*, *NKX6.1*, *NGN3* and *SOX9* (Fig. 4c). Gene network analysis highlighted that significantly downregulated categories were enriched with factors involved in pancreatic secretion, insulin secretion and MODY (Fig. 4e). Furthermore, RNA-seq analysis revealed upregulation of genes encoding non-canonical WNT ligands, such as *WNT5A*, *WNT5B*, *WNT7A*, in GLI2^{P>L HET} endocrine progenitors (Fig. 4b, d and Supplementary Fig. 4b, d). RT-qPCR further validated the dysregulation of WNT ligands (*e.g.*, WNT2, WNT5A, WNT7A) and WNT receptors (*e.g.*, FZD3, FZD7, FZD8) as well as GLI transcripts and components of the HH pathway in GLI2^{P>L HET} endocrine progenitor cells (Fig. 4f). Interestingly, the transcript levels of *GLI2* and *GLI3* were upregulated in GLI2^{P>L HET} starting from endocrine progenitor stage, while *GLI1* expression was downregulated (Supplementary Figs. 2i, 4b; Supplementary Data 4, 5). This could be the result of an attempt of compensating HH downstream inactivation.

Recent observations pointed out to multiple distinct roles of the non-canonical WNT pathway in endoderm and endocrine cell differentiation, morphogenesis, and maturation, entailing its tight regulation during pancreatic development[34–37].

Because of the significant upregulation of *WNT5a* in GLI2^{P>L HET} iPSCs from gut tube stage onward (Fig. 4b, d, f; Supplementary Fig. 4b, d, e), we hypothesized that its aberrant and sustained activation may contribute to impaired endocrine cell development in the mutant lines. To test this hypothesis, we either stimulated or inhibited the non-canonical WNT signaling pathway in GLI2^{CTRL} and GLI2^{P>L HET} iPSCs, respectively (Fig. 5a). GLI2^{CTRL} cells were treated with WNT5A for 4 days after the induction of *PDX1* and acquisition of pancreatic fate, and, subsequently, analyzed by RT-qPCR. After exposure to WNT5A, we observed a decrease in the expression of pancreatic (*PDX1*, *NKX6.1*) and endocrine (*NGN3*, *NKX2.2*, *NEUROD1*) gene markers in GLI2^{CTRL}-derive

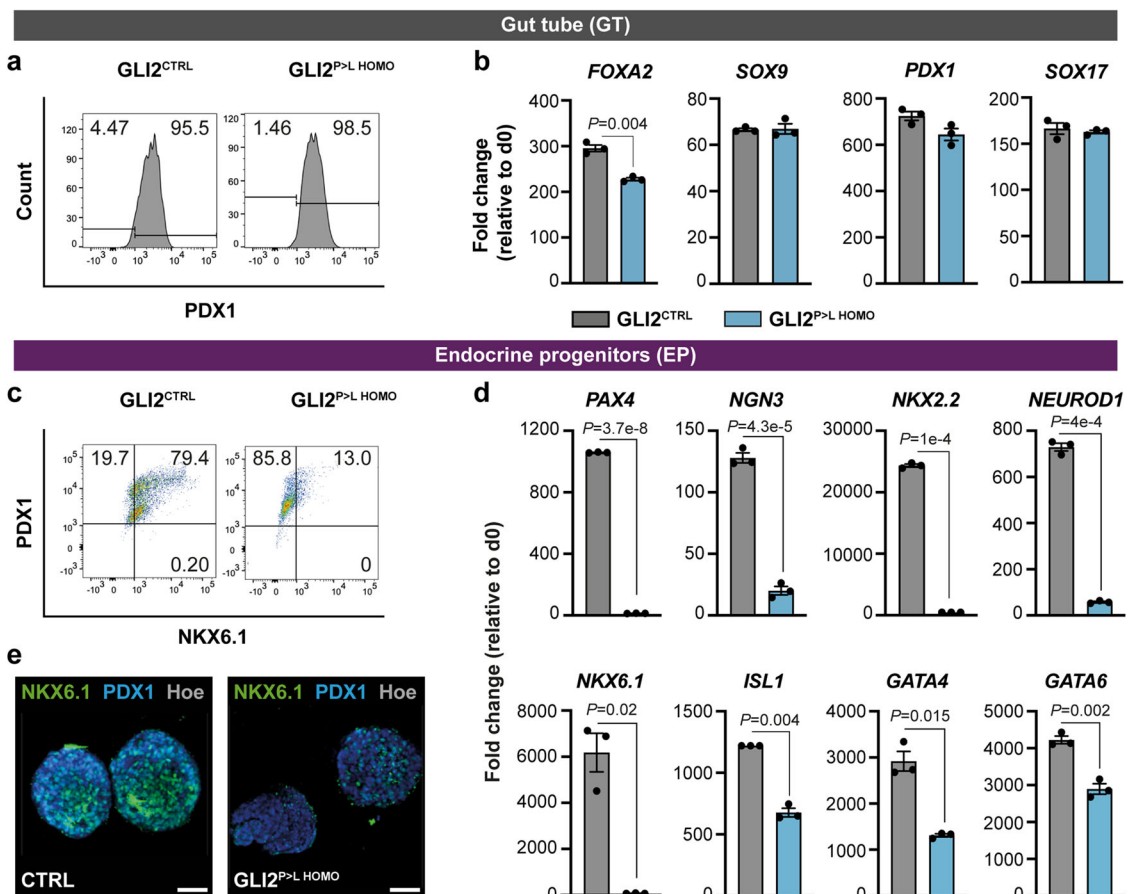

**Fig. 3 | Characterization of homozygous GLI2$^{P>L}$ HOMO cells during pancreatic cell differentiation. a** Representative flow cytometry plots of PDX1$^+$ cells (shown as %) in GLI2$^{CTRL}$ and GLI2$^{P>L HOMO}$-derived cells at day (D) 5 of differentiation. $n = 3$ differentiation experiments on two GLI2$^{P>L HOMO}$ independent clones. **b** RT-qPCR analysis of selected gene transcripts in GLI2$^{CTRL}$- and GLI2$^{P>LHOMO}$ differentiated cells at D5. Values were normalized to *GAPDH* and shown as fold change relative to undifferentiated cells (D0). Values shown are mean ± SEM. $n = 3$ differentiation experiments on two GLI2$^{P>LHOMO}$ independent clones. **c** Representative FACS plot of NKX6.1$^+$ and PDX1$^+$ cells (shown as %) in GLI2$^{CTRL}$- and GLI2$^{P>L HOMO}$-derived cells at

D14. $n = 3$. **d** RT-qPCR analysis of selected gene transcripts in GLI2$^{CTRL}$ and GLI2$^{P>L HOMO}$ differentiated cells at D14. Values were normalized to *GAPDH* and shown as fold change relative to undifferentiated cells (D0). Values shown are mean ± SEM. $n = 3$. Two-tailed Student's $t$-test; exact $P$-values are reported in the figure. Source data are provided as a Source Data file. **e** Whole-mount immunostaining for PDX1 and NKX6.1 in GLI2$^{CTRL}$ and GLI2$^{P>L HOMO}$-derived endocrine progenitor clusters. Immunostaining experiments were performed 2 independent times, and each experiment showed similar results. Nuclei were labeled with Hoechst (Hoe). Scale bars, 50 μm.

endocrine progenitors (Fig. 5b). RT-qPCR analysis also showed upregulation of *WNT5A* itself and its downstream targets, *LAMC2* and *TEAD4*, in stimulated GLI2$^{CTRL}$ cells (Fig. 5b). GLI2$^{CTRL}$ cells that were further differentiated to β-like cell stage after exposure to WNT5A showed a decrease in the expression levels of *NKX2.2* and *NEUROD1*, accompanied by a reduction of gene transcripts essential for human β-cell functionality, such as *PCSK1*, as well as endocrine hormones, including *INSULIN*, *GLUCAGON*, *SOMATOSTATIN* (Fig. 5c). Exposure to WNT5A also affected the functional maturation of the cells, which showed reduced glucose-stimulated insulin secreting (GSIS) capacity as compared to GLI2$^{CTRL}$-derived β-like cells, but similar to GLI2$^{P>L HET}$ mutant cells (Fig. 5d). Together, these results demonstrate that inappropriate activation of WNT5A signaling impairs differentiation of iPSCs into β-like cells and recapitulates the phenotype of GLI2$^{P>L HET}$ mutant cells.

We next investigated if inhibition of WNT5A could instead rescue β-cell development in GLI2$^{P>L HET}$ iPSCs. Cells were treated with BOX5, a WNT5A antagonist, which was previously shown to attenuate WNT5A-mediated Ca$^{2+}$ and protein kinase C signaling[38] (Fig. 5a). GLI2$^{P>L HET}$ clusters treated with BOX5 exhibited a significant increase in the expression of *PDX1*, *NKX6.1* and other endocrine markers (Fig. 5b). Moreover, the expression of *WNT5A* and its targets was reduced after addition of BOX5 (Fig. 5b), confirming the activity of the compound.

Later at D21, BOX5-treated GLI2$^{P>L HET}$-derived β-like cells restored the expression of β-cell genes (*NKX6.1*, *NKX2.2*, *NEUROD1*, *PCSK1*, *INSULIN*) to levels comparable with control cells. Consistently, we found that exposure to BOX5 ameliorates the functional defects of GLI2$^{P>L HET}$-derived β-like cells, with the treated cells showing an increase of insulin release in response to glucose stimulation (Fig. 5d). Overall, these findings indicate that WNT5A signaling regulation is critical for pancreatic progenitor development and its inhibition partially rescues the GLI2$^{P>L HET}$ defects that may ultimately contribute to diabetes onset.

## Discussion

We report a previously uncharacterized heterozygous missense variant p.P1554L in the *GLI2* gene in patients with early-onset diabetes of unknown origin. Isogenic iPSC lines with either a heterozygous or homozygous p.P1554L mutation in the *GLI2* gene showed impaired pancreatic progenitor and endocrine development in a dose-dependent manner, suggesting haploinsufficiency as an underlying mechanism of disease. The iPSC models have allowed us to move beyond the discovery of the mutant variant to an understanding of the causative variant and the molecular mechanisms of the disease. Our findings suggest that the diminished expression of *PDX1* and *NKX6.1* in patient-like iPSCs from pancreatic progenitor stage onwards is responsible for the endocrine progenitor pool depletion, which

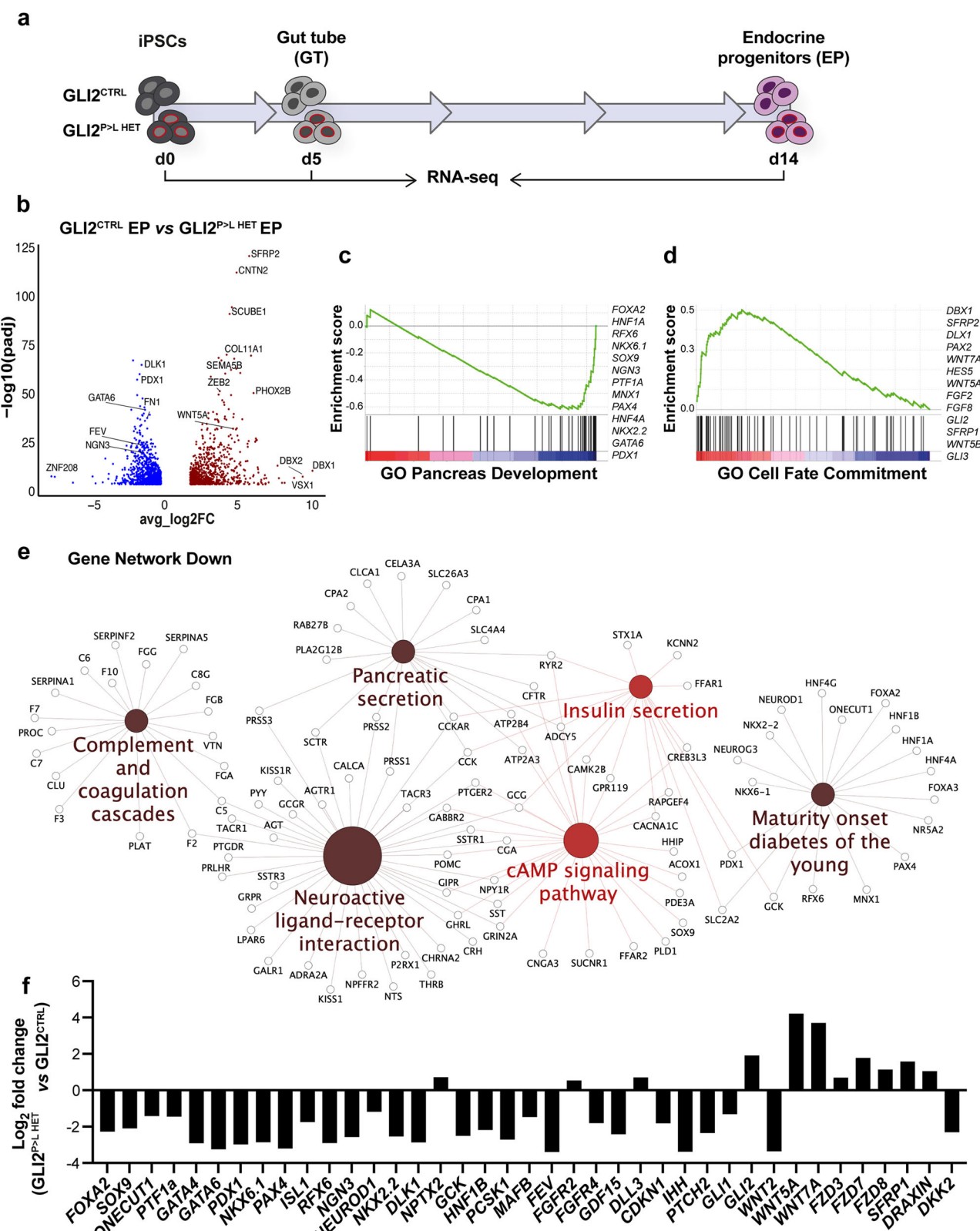

ultimately results in reduced β-cell mass at birth and explain the onset of the diabetes in young age. Likewise, heterozygous mutations in other transcription factors (*e.g., GATA6, HNF1B*) have been reported to cause diabetes as a consequence of their effects on early pancreatic development[4,9,10,39–41]. Heterozygous carriers these mutations often have incomplete penetrance and variable onset of disease, ranging from pancreatic agenesis to adult-onset diabetes[4,10,39,41,42]. Marked

phenotypic variability was observed also among individuals with the identified GLI2 p.P1554L variant within the same family, ranging from severely to mildly diabetic developed in adulthood, to non-diabetic at the time of the investigation. This suggests that other genetic or environmental factors might modify the phenotype and influence the development of disease. Further studies are also required to overcome the limitations and concerns which remain around the ability of exome

**Fig. 4 | Whole-transcriptome analysis of GLI2$^{P>L}$ HET iPSCs undergoing differentiation into endocrine progenitor cells. a** Schematic of the experimental design. RNA-seq data were obtained from GLI2$^{CTRL}$ and GLI2$^{P>L HET}$ iPSCs at day (D) 0 (undifferentiated stage) and undergoing differentiation at D5 (gut tube stage) and D14 [endocrine progenitor (EP)] stage from two independent biological replicates. **b** Volcano plots visualizing the global transcriptional change across the groups compared (D14 GLI2$^{CTRL}$ *vs* GLI2$^{P>L HET}$). Each data point in the plot represents a gene. Genes with an adjusted *P*-value <0.05 and a log2 fold change >1 are indicated by red dots. These represent upregulated genes. Genes with an adjusted *P*-value <0.05 and a log2 fold change <−1 are indicated by blue dots. These represent downregulated genes. Wald test was used as statistical test as implemented in DESeq2 package (see Methods). **c, d** Gene set enrichment analysis (GSEA) plots of representative gene set downregulated (**c**) or upregulated (**d**) in GLI2$^{P>L HET}$ *vs* GLI2$^{CTRL}$ EPs at D14. GSEA *P*-values are derived from permutation test and corrected for multiple testing using

the FDR method. Nominal *P*-value = 0.00, FDR = 0.00. **e** GO term enrichment and pathway term network analysis of DEGs between GLI2$^{CTRL}$- and GLI2$^{P>L HET}$-derived EPs. Gene Network showing downregulated genes (FDR ≤ 0.05). Term node size increases with term significance and color corresponds to a particular functional group that is based on the similarity of their associated genes. Groups that shared 50% or more of the same genes were merged. The proportion of each node that is filled with color reflects the kappa score. Functionally related groups partially overlap. **f** RT-qPCR validation of a subset of DEGs in GLI2$^{CTRL}$- and GLI2$^{P>L HET}$-derived EPs. All tested genes showed concordant differential expression with RNASeq results. Validation was performed on an independent differentiation experiment. Data were normalized to that of GAPDH and shown as Log2-expression ratio between GLI2$^{P>L HET}$ and GLI2$^{CTRL}$ EP cells. Source data are provided as a Source Data file.

sequencing to provide adequate coverage for all genes, exons, and variants as well as the types of variants it can detect[5]. Indeed, exome sequencing does not inform on variants outside exons (*e.g.*, on promoters, enhancers) that can affect gene regulation[5]. Additionally, the highly heterogeneous phenotype presented by the different carriers in the family could be the result of copy number variation and/or structural variants that have not been examined here. Lastly, the phenotypic heterogeneity could be explained by a compound heterozygosity model, whereby in the two very early-onset diabetic siblings the GLI2 p.P1554L variant might be present in combination with a second currently unknown variant also affecting GLI2. Future genetic studies of the family may further elucidate the genetic mechanism underpinning the observed range of disease phenotypes.

Precise spatiotemporal HH regulation is crucial for pancreas development and β-cell function in mice[17–19,43]. Mesenchymal HH signaling has been shown to play the major role during pancreas development and pancreatic cancer, while the developing pancreatic epithelium appeared less sensitive to deregulation of HH signaling[19,44]. Here, we uncovered a role for *GLI2* in human pancreatic development. We found that *GLI2* is expressed in a subset of pancreatic progenitors, before endocrine specification, and its expression persists at later stages in human islet cells. Together, our results support that the p.P1554L variant, which lies within the TAD of GLI2, impairs its transcriptional activity, possibly through changes in partner binding, as previously reported in other contexts[45,46]. GLI-dependent luciferase reporter activity was indeed significantly reduced in HEK293T cells transfected with GLI2p.P1554L variant as well as in GLI2$^{P>L HET}$ iPSCs in comparison to GLI2$^{CTRL}$ cells. Additionally, our RNASeq analysis showed the dysregulation of a set of HH pathway components in GLI2$^{P>L HET}$ pancreatic cells, including downregulation of *PTCH2*, *IHH*, *SHH*, *GLI1* and upregulation of *GLI3*, which can act as a HH inhibitor[15,46]. Thus, the fine balance between the GLI factors (activator and repressor), which is typically at play to regulate the HH pathway[15,16,46], seems here to result into an overall negative signal output. Consistently, the increased levels of *GLI2* transcript observed in GLI2$^{P>L HET}$ pancreatic cells, starting at endocrine progenitor stage, suggest an attempt to compensate HH inhibition in the mutant cells.

RNASeq and network analyses also revealed a crosstalk between HH/GLI2 and WNT pathways in human pancreatic progenitors, which results into dysregulation of non-canonical WNT signaling in GLI2$^{P>L HET}$ mutant cells. Notably, rescue experiments (by blocking WNT5A) as well as mimicking experiments (by adding WNT5A) showed that GLI2 functions in endocrine progenitors through mechanisms that regulate WNT signaling. A crosstalk between HH/GLI and WNT pathways has been reported in other contexts, such as cancer[47], neurogenesis[48] and differentiation[49]. However, the precise mechanisms that regulate HH and WNT activity seem to be highly context-specific and the events by which changes in HH/GLI2 signaling in pancreatic cells determines WNT regulation remain to be fully elucidated. Stage-specific WNT signaling regulates multiple aspects of pancreas development in

different species, including patterning, specification, and differentiation of pancreatic cells[34,36,50–52]. Recent studies reported a non-cell autonomous WNT5A-mediated role for β-cell differentiation from endocrine progenitors in mouse[53] and human stem cell models[35]. By contrast, WNT5A/PCP role in β-cell maturation is still not fully understood and might be different between mouse and human[34,54]. Moreover, *WNT5A* is expressed in the pancreatic epithelial component, beside the mesenchyme, and it is unclear how the expression of this ligand is coordinated in the epithelial/mesenchyme compartment to regulate endocrine cell formation. Here, we identified an additional stage-specific response to cell autonomous WNT that corresponds to the time when human pancreatic progenitors acquire endocrine identity. At this specific stage, it is likely that the crosstalk between the two pathways confers temporal control to WNT, wherein GLI2 activation might sustain canonical WNT and prevent aberrant non-canonical WNT signaling during the acquisition of human endocrine progenitor fate. The diverse roles and mechanisms regulating HH and WNT crosstalk during pancreatic lineage differentiation remain to be fully elucidated. Also, while we have presented evidence for HH regulation of non-canonical WNT, the mechanism by which GLI2 affects WNT responsiveness is not known in pancreatic progenitors and is a subject for future research. Overall, such knowledge could be harnessed to improve current targeted differentiation protocols for optimizing β-cell differentiation from human PSCs.

## Methods
### Human material and patient variant discovery
The index child was part of a cohort of 94 individuals recruited from Charité Medical School and international diabetes centers[55]. All individuals with hyperglycemia tested negative for β-cell autoantibodies and developed diabetes before the age of 18 years. Mutations in known genes causing monogenic diabetes (*GCK*, *HNF4A*, *HNF1A*, *HNF1B*, ABCC8, *KCNJ11*, and *INS*) were excluded by Sanger sequencing. We performed exome sequencing in the child with diabetes and his consanguineous parents and sibling using the Agilent SureSelect Human All Exon Kit (Agilent SureSelect v4, 50 Mb) and next-generation sequencing (Hiseq, Illumina, USA). We analyzed the data with our established pipeline[55,56] and confirmed the GLI2 variant (p.P1554L) by Sanger sequencing (Applied Biosystems, USA). The Charité committee on human research approved the study (EA-No EA2/054/11) and written informed consent to publish clinical information was obtained from all participants at enrollment.

The human embryonic and fetal material was provided by the joint MRC/Wellcome Trust (grant# MR/X008304/1 and 226202/Z/22/Z) Human Developmental Biology Resource (http://hdbr.org) with appropriate maternal written consent and approval from the Newcastle and North Tyneside NHS Health Authority Joint Ethics Committee (23/NE/0135) and London Fulham Research Ethics Committee (23/LO/0312). The HDBR is regulated by the UK Human Tissue Authority (HTA; www.hta.gov.uk) and operates in accordance with the

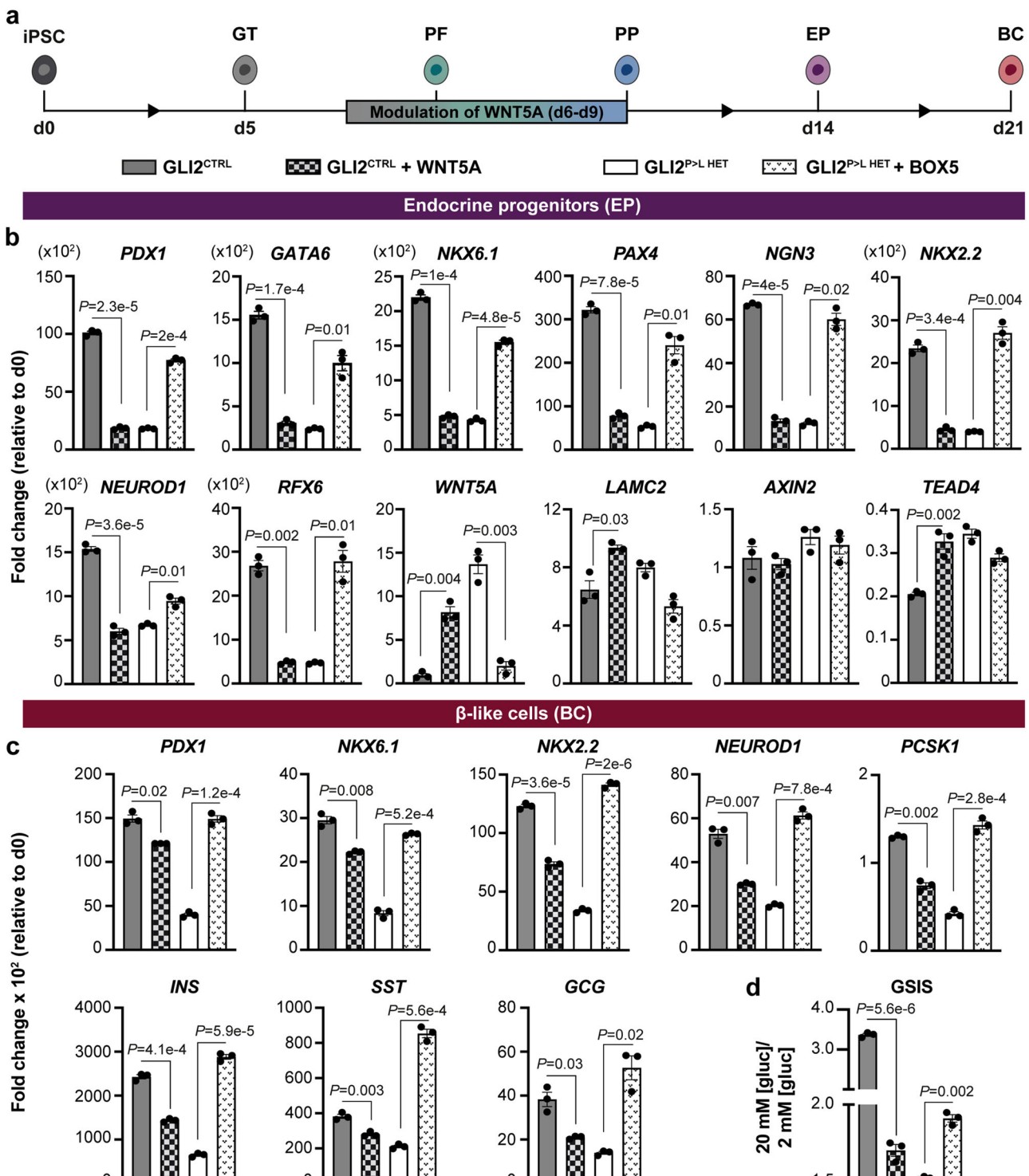

**Fig. 5 | Modulation of WNT5A signaling rescues β-cell differentiation in GLI2[P>L] HET iPSCs. a** Schematic representation of the differentiation protocol of iPSCs into β-like cells. Cell clusters were treated in suspension with WNT5A recombinant protein or BOX5 for 4 days at pancreatic progenitor (PP) stage of differentiation. **b** RT-qPCR analysis of selected gene transcripts in GLI2[CTRL]- and GLI2[P>L HET]-derived endocrine progenitors (EP) at day (D) 14 after the indicated treatments. Data are represented as fold change relative to undifferentiated cells (D0). Values shown are mean ± SEM. *n* = 3 differentiation experiments. Two-tailed Student's *t*-test; exact *P*-values are reported in the figure. **c** RT-qPCR analysis of selected gene transcripts in GLI2[CTRL]- and GLI2[P>L HET] differentiated cells at D21 after the indicated treatments. Data are represented as fold change relative to undifferentiated cells (D0). Values shown are mean ± SEM. *n* = 3 differentiation experiments. Two-tailed Student's *t*-test; exact *P*-values are reported in the figure. **d** Representative static GSIS assay of D21 GLI2[CTRL]- and GLI2[P>L HET]-derived β-like cells after the indicated treatments. On *Y*-axis ratio of insulin secreted at high *versus* low glucose [Glc] conditions. Values shown are mean ± SD. *n* = 3 differentiation experiments. Two-tailed Student's *t*-test; exact *P*-values are reported in the figure. Source data are provided as a Source Data file.

relevant HTA Codes of Practice. Fixed human embryonic pancreatic tissue samples (gender not established) were provided following overnight fixation in 4% paraformaldehyde (PFA). The tissue samples were then processed for cryosectioning, immunostaining and imaging at King's College London, UK. All work was undertaken in approval of the HDBR Steering Committee to the Spagnoli lab. at King's College London, UK (License #200523).

## Cell Lines and Cell Culture

HEK293T cells were purchased from ATCC (CRL-3216) and cultured in DMEM medium supplemented with 10% FBS and 1% Pen/Strep at 37 °C, 5% $CO_2$. Human iPS cell line HMGUi001-A2 (sex: female), a sub-clone of HMGUi001-A[57], was kindly provided by Dr Kühn (MDC) and authenticated by karyotyping. Human iPSCs were maintained on Geltrex-coated (Invitrogen) plates in E8 media (STEMCELL Technologies). The medium was changed daily, and cells were passaged every ~3 days as cell clumps or single cells using 0.5 mM EDTA (Invitrogen) or Accutase (Invitrogen), respectively. Medium was supplemented with 10 μM Rho-associated protein kinase (ROCK) inhibitor Y-27632 (Sigma) when iPSCs were thawed or passaged as single cells. All cell lines tested negative for mycoplasma contamination which was carried out routinely.

For transient transfection experiments in HEK293T cells were transfected using Polyethylenimine "Max" (PEI) (Polysciences) in OptiMEM medium (Invitrogen).

## Site-directed mutagenesis

Site-directed mutagenesis was used to introduce the patient variant GLI2 c.C4661T into the pCS2-MT GLI2 FL plasmid. Mutagenesis was performed using the QuickChange II XL Site-Directed Mutagenesis Kit (Agilent) following manufacturer's guidelines. Mutation was generated using the following specific primer pair:

GLI2 c.C4661T-F: GCGGGGATGGAGAGCAGGGTCAAGG
GLI2 c.C4661T-R: CCTTGACCCTGCTCTCCATCCCCGC

## GLI-responsive luciferase assay

For luciferase assays, the dual luciferase reporter assay kit (Promega) was used. HEK293T or pluripotent iPSC cells were seeded at $1.3 \times 10^6$ cells in 12-well plates and transiently co-transfected in duplicates with 0.5 μg of wild-type or mutant GLI2 pCS2-MT plasmids together with 0.5 μg of GLI-responsive Firefly luciferase reporter construct (8 × 3'Gli-BSδ51LucII) and 0.05 μg of a constitutive Renilla luciferase reporter (pRL-SV40) construct. Protein lysates were prepared 48 hrs after transfection. Firefly and Renilla Luciferase activities were quantified using the dual reporter assay kit (Promega) according to the manufacturer's instructions on an Infinite 200 Pro-luminometer (TECAN). Luciferase assay experiments were repeated three times on independent samples. As negative controls, the mutated version of the GLI-luciferase reporter construct (8xm3'Gli-BSδ51LucII) was used, as well as a GFP-expressing construct. Firefly/Renilla activity ratio was then calculated for each sample.

## Differentiation of pluripotent iPSCs into pancreatic β-like cells

Differentiation was carried out following a 21 day protocol previously described by Russ et al.[29]. Briefly, iPSCs were dissociated using Accutase and seeded at a density of $5.5 \times 10^6$ cells per well in ultra-low attachment 6-well plates (Thermo Fisher Scientific) in E8 medium supplemented with 10 μM ROCK inhibitor, 10 ng/ml Activin A (R&D Systems) and 10 ng/ml Heregulin (Peprotech). Plates were placed on an orbital shaker at 100 rpm to induce sphere formation at 37 °C in a humidified atmosphere containing 5% $CO_2$. To induce definitive endoderm differentiation, cell clusters were incubated in Day (D)1 medium [RPMI (Invitrogen) containing 0.2% FBS, 1:5000 ITS (Invitrogen), 100 ng/ml Activin A, 50 ng/ml WNT3a (R&D Systems)] into low attachment plates. Subsequently cell clusters were differentiated into

β-like cells by exposure to the appropriate media as previously published[29]. All recombinant proteins were purchased from R&D System unless otherwise stated (see Supplementary Data 2).

For static GSIS assays, D21 cells (eight to ten clusters, equivalent to ~0.5 − 1.0 × 10^6 cells) were rinsed twice with Krebs buffer (129 mM NaCl, 4.8 mM KCl, 2.5 mM CaCl₂, 1.2 mM MgSO₂, 1 mM Na₂HPO₄, 1.2 mM KH₂PO₄, 5 mM NaHCO₃, 10 mM HEPES, 0.1% BSA, in deionized water and then sterile filtered) and then pre-incubated in Krebs buffer for 60 min. After preincubation, the cells were incubated in Krebs buffer with 2 mM glucose for 60 min (first challenge) and supernatant collected. Then clusters were washed once in Krebs buffer, transferred onto another plate containing fresh Krebs buffer with 20 mM glucose (second challenge) and incubated for additional 60 min. Supernatant samples were collected after each incubation period and frozen at −70 °C for human C-peptide ELISA (cat. #10−1141-01; Mercodia) and Insulin (cat. #10−1132-01; Mercodia) measurement.

## Generation of clonal HMGUi001-A2 patient-like mutant lines

Patient-like isogenic iPSC lines were generated using the doxycycline (DOX)-inducible iCRISPR system previously established by Yumlu et al.[27]. Briefly, the DOX-inducible promoter TREtight and the Cas9 gene were knocked-in between the first two exons of the PPP1R12C gene (AAVS1 locus) on one allele. The other allele harbored the Transactivator gene rtTA3 under the constitutively active CAGGS promoter. Addition of DOX mediates binding of rtTA to TREtight, thus activating the expression of Cas9. After transfection of iPSCs with a plasmid carrying the sgRNA together with a fluorescent reporter (Venus) and a repair template (single stranded oligo-dinucleotide [ssODN]), efficient precise knock-in can be introduced by HDR upon DOX-induction of Cas9 (Supplementary Fig. 1). sgRNAs targeting the site of mutation were designed using the CRISPOR website (http://crispor.tefor.net/) and cloned into the pU6-(BbsI)sgRNA_CAG-venus-bpA vector by BbsI restriction enzyme overhangs. The sgRNA expression vector was a kind gift of Dr Kühn (MDC) (Addgene #86986). The respective point mutation was introduced through HDR using a 119 bp long ssODN. One day prior to transfection cells were seeded at $0.2 \times 10^6$ cells per well of pre-coated 6-well plate as single cells. For targeting, cells were transfected with 0.5 μg sgRNA vector and 30 pmol ssODN using Lipofectamine 3000 (Invitrogen) according to manufacturer's instructions. The next 2 days cells were fed with medium supplemented with 1 μg/ml DOX and then live-sorted for Venus. After FACS, single cells were expanded for 3 weeks to form colonies that were screened for recombination with the donor template. Positive clones were validated by Sanger sequencing. Sequences of sgRNA, ssODN and genotyping primers are listed in Supplementary Data 1. Two heterozygous and two homozygous mutant HMGUi001-A2 iPSC lines carrying the c.C4661T point mutation in GLI2 were established and compared to the isogenic control iPSC line for their pluripotency and differentiation properties.

## Cell sorting and flow cytometric analysis

Transfected iPSCs or differentiated cell clusters were dissociated with Accutase (Invitrogen). For sorting, cell suspension was filtered and resuspended in PSB with ROCK inhibitor (Sigma). Fluorescent Activated Cell Sorting (FACS) for Venus-expressing cells was performed on FACS Aria I or II (BD Bioscience). Sorted iPSCs were collected in E8 medium with ROCK inhibitor (Sigma) and seeded at low density to derive single-cell colonies. For flow-based analysis, dissociated cell clusters were fixed for 20 min at 4 °C in cold BD fixation/permeabilization™ solution (BD Bioscience). Cells were washed twice in BD Perm/Wash™ Buffer (BD Bioscience) and incubated with primary antibodies in the dark for 2 hrs at 4 °C. Cells were washed 3x in BD Wash™ Buffer (BD Bioscience), resuspended in BD-FBS staining™ buffer (BD Bioscience) with secondary antibodies and incubated for 1 hr at RT. Primary and secondary antibodies are listed in the

Supplementary Data 2 and used at a concentration of 1:100. Stained cells were washed 3x and analysis was performed on FACS Aria II. FlowJo v10 software was used to analyze data. For gating, samples unstained were used as negative controls. Representative flow cytometry pseudocolor plots and gating strategy are shown in Supplementary Fig. 5.

## Immunofluorescence staining

For immunofluorescence, iPSCs were grown on glass coverslips. Cells were fixed for 20 min with 4% PFA at RT and washed 3x with 1 x PBS. Cells were blocked for 30 min in 0.1% Triton-PBS containing 3% donkey serum. Incubation with primary antibody was performed overnight at 4 °C in blocking solution. The next day cells were washed 3x for 10 min each in 0.1% Tween- PBS, incubated for 1 hr at RT in secondary antibody solution (in blocking buffer) and washed as above. Coverslips were mounted on slides using in DAKO® fluorescence mounting medium. The antibodies and dilutions are listed in Supplementary Data 2.

iPSC-derived cell clusters were fixed in 4% PFA for 15 min at RT and stained either as whole-mounts or embedded for cryosectioning. For histological analysis after fixation, cell clusters were washed in PBS and equilibrated in 15% sucrose solution (PBS) at RT. Cell clusters were first embedded in 2.5% low-melting agarose, then in O.C.T. compound (Tissue-TekÒ, SakuraÒ, Finetek) and subsequently frozen and stored at −80 °C. Cryosections were cut at a thickness of 10 μm using a CM3050 S Leica cryostat and collected on standard glass slides (Thermo Scientific). Before immunostaining, slides were incubated at 37 °C for 45 min to remove agarose. Cluster sections were circled with a PAP pen to provide a hydrophobic barrier for application of solutions. Next, sections were blocked for 30 min at RT with 0.1% Triton-PBS containing 3% donkey serum. Primary antibodies were diluted according to Supplementary Data 2 in blocking buffer and sections were incubated overnight at 4 °C in primary antibody solution. The next day, slides were washed 3x for 5 min each in PBS with 0.1% Tween. Secondary antibodies and Hoechst were diluted 1:750 in blocking buffer. Slides were incubated for 40 min with secondary antibody conjugated to Alexa fluorophores in blocking buffer. Before imaging, slides were washed in PBS + 0.1% Tween and mounted in DAKO® fluorescence mounting medium. Images were acquired on a Zeiss LSM 700 confocal microscope using a 40x oil immersion objective.

## RNA isolation, reverse transcription and quantitative PCR

The High Pure RNA Isolation Kit (Roche) was used for RNA extraction from cultured cells. Total RNA was processed for reverse transcription (RT) using Transcriptor First Strand cDNA Synthesis Kit (Roche). Real-time PCR reactions were carried out using SYBRGreen Master Mix (Roche) on LightCycler 96 system (Roche). Human glyceraldehyde 3-phosphate dehydrogenase (GAPDH) was used as reference gene. Mouse and Human Primer sequences are provided in Supplementary Data 1. Gene expression levels were determined by the $2 - \Delta \Delta CT$ method following normalization to reference genes. RT-qPCR experiments were repeated at least three times with independent biological samples; technical triplicates were run for all samples; minus RT and no template controls were included in all experiments.

For RNA sequencing (RNA-seq), total RNA from GLI2$^{CTRL}$ and GLI2$^{P>L\ HET}$ cells at iPSC undifferentiated, gut tube and endocrine progenitor stage were extracted from two independent biological replicates. Total RNA concentration and RNA integrity of each sample were determined with NanoDrop and Qubit 4 Fluorometer (Thermo Fisher Scientific). RNASeq library preparations and sequencing reactions were conducted by GENEWIZ.

## Bioinformatics

Initial bioinformatics analysis of the RNASeq was conducted by GENEWIZ. Briefly, data was generated with an Illumina HiSeq 2 x 150 PE HO

configuration. Sequence reads were trimmed to remove adapter sequences and nucleotides with poor quality (Trimmomatic v.0.36). Using the STAR aligner v.2.5.2b the trimmed reads were mapped to the Homo sapiens GRCh38p13 reference genome available on ENSEMBL. Gene expression between distinct groups were compared using DESeq2 package. The Wald test was used to generate P-values and log2 fold changes. Genes with an adjusted P-value < 0.05 and absolute log2 fold change > 1 were called as differentially expressed genes for each comparison. Gene Ontology analysis was performed using the publicly available Database for Annotation, Visualization, and Integrated Discovery (DAVID) bioinformatics tool https://david.ncifcrf.gov.home.jsp. The required data for the DEGs were uploaded to DAVID via the "RDAVIDWebService" BioConductor library, where each DEG was assigned to relevant GO terms with subsequent selection of significantly enriched GO terms from the GO BP Direct database.

For the GSEA analysis, the GSEA software package (Desktop v4.1.0) developed by the MIT/BROAD Institute was used[58,59]. Gene sets M15945 (GOBP_CELL_FATE_COMMITMENT) and M11915 (GOBP_PANCREAS_DEVELOPMENT) were used for enrichment of pancreas development-related genes. All gene set files for this analysis were obtained from GSEA website www.broadinstitute.org/gsea/. Enrichment map was used for visualization of the GSEA results. GSEA computes four key statistics for the gene set enrichment analysis report: https://www.gsea-msigdb.org/gsea/doc/GSEAUserGuideTEXT.htm#_GSEA_Statistics. GSEA P-values were derived from permutation testing and corrected for multiple testing using the False discovery rate (FDR) method. Enrichment score (ES) and FDR value were applied to sort cell fate- and pancreatic development-enriched after gene set permutations were performed 1000 times for the analysis. Gene Network Analysis was based on DEGs (log2foldchange > 1) obtained when comparing gene expression at each of days 0, 5 and 14 during pancreatic differentiation between GLI2$^{CTRL}$ and GLI2$^{P>L\ HET}$ cells. Enrichment of Kyoto Encyclopedia of Genes and Genomes (KEGG) ontology terms and pathways and gene network construction was conducted in Cytoscape[60] (v3.8.0) using the ClueGO (v2.3.6) and Cluepedia (v1.3.6) plugins. KEGG enrichment of proteins in the following section was carried out with the R package cluster Profiler that enables enrichment comparisons across multiple clusters of genes/proteins. All programming code was generated with the aid of package vignettes made accessible via The Comprehensive R Archive Network (CRAN).

## Statistics and reproducibility

Statistical analyses were performed using GraphPad Prism. Unless stated otherwise, data are shown as mean ± standard error of the mean (SEM) and statistical significance was determined using Two-tailed Student's t-test; exact P-values are reported in the figures.

The detail of biological replicates is summarized below or included in the Figure legends. In Fig. 1e, immunofluorescence experiments were performed two independent times per embryonic stage, and each experiment showed similar results. In Fig. 3e, whole-mount immunofluorescence experiments were performed 3 independent times, and each experiment showed similar results.

## Reporting summary

Further information on research design is available in the Nature Portfolio Reporting Summary linked to this article.

## Data availability

The RNA-sequencing data of the human iPSC lines (Control and GLI2$^{P>LHET}$ mutant) generated in this study have been deposited in the GEO database under the accession code GSE224943 https://www.ncbi.nlm.nih.gov/geo/query/acc.cgi?acc=GSM7035978 and are publicly available. No original code was reported in this study. Raw data to generate all graphs within the Figures and Supplementary Figures. are provided as a Source Data File. Source data are provided with this paper.

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

## Acknowledgements

We acknowledge the support of the EFSD Research Grant [grant number 1117775] to F.M.S. and the Transcard Helmholtz PhD studentship to L.M. A.I. is the recipient of a Wellcome Trust PhD studentship [grant number 222354/Z/21/Z] and A.S. of a Wellcome Trust studentship from the PhD program 'Advanced therapies for regenerative medicine' [grant number 218461/Z/19/Z]. K.R. was supported by the ECRC and Berlin Institute of Health (BIH). We appreciate all the studied individuals. We thank the Human Developmental Biology Resource (http://hdbr.org) for the help in collecting the pancreatic fetal tissue. We thank the support of the BIH / MDC Pluripotent Stem Cells Core Facility, Dr S. Diecke and Dr R. Kuhn at the MDC for advice on iPSC CRISPR targeting at the beginning of the study. We are grateful to the BRC Flow Cytometry platform at King's College London.

## Author contributions

F.M.S. and L.M. designed the study, conceived the experiments, and wrote the manuscript with input from remaining authors. K.R. and N.S.E. conducted and coordinated the clinical study. M.G. performed the sequencing experiments. L.M., A.I., A.S., I.T. and H.W. performed and analyzed all the experiments in human iPSCs.

## Competing interests

The authors declare no competing interests.
