## [Peer Review File · Nature Communications]

Heterozygous missense variant in *GLI2* impairs human endocrine pancreas developmentREVIEWER COMMENTS

Reviewer #1 (Remarks to the Author):

The manuscript submitted by Mueller et al. presents a novel analysis of a missense mutation in the Hedgehog signalling transcription factor GLI2 that was first observed in clinical diagnosis of diabetes with an unknown cause. The authors link the presence of GLI2-associated activity to endocrine and beta cell commitment using a GLI2-mutated human induced pluripotent stem cell line with CRISPR-editing. To investigate downstream mechanisms underlying the impairment in pancreatic differentiation the authors perform RNAsequencing at various stages of differentiation and uncover differences in the expression levels of non-canonical wnt ligands and further demonstrate that inhibition of non-canonical wnt could rescue pancreatic differentiation in the iPSC line carrying the missense mutation. The data are well presented. I am unsure however about the developmental stage impacted by the mutation. The authors suggest that GLI2 mutation impairs endocrine progenitor development, I would argue that the impairment occurs earlier during the formation of the pancreatic progenitor cells, as the authors detect lower NKX6-1 expression already between d9-d11 of differentiation (Fig 2c and suppl fig 3b). I would expect a lower frequency of PDX1+/NKX6-1+ cells already at the pancreatic progenitor stage which would manifest in fewer endocrine progenitor cells at day 14. Their rescue experiment is consistent with this idea, as they modulate non-canonical wnt5a prior to pancreatic progenitor differentiation rather than during endocrine progenitor specification.

Suggestions to strengthen the manuscript:

- 1) A major question that is not clear throughout the text is the role of GLI2 in Hedgehog signalling and how the Hedgehog signalling axis is affected with the GLI2 P>L mutation. The authors do show a GLI responsive luciferase assay, but they fail to comment on possible compensation or modulation through GLI1 or GLI3 in GLI2 P>L cells. There is interesting information provided in the RNA sequencing data provided in Fig. 3 and Supp. Fig. 4 that suggests down-regulation of GLI1 and up-regulation of GLI2 and GLI3. The authors do comment on page 5 of the manuscript regarding the transcriptional activator or repressor roles of GLI, but any further characterization of what is happening to Hedgehog signalling in GLI2 P>L cells is lacking. Is SANT-1 required during the differentiation of the mutant lines? What happens in the absence of SANT-1? The authors could also benefit from adding more information regarding Hedgehog signalling in pancreatic and islet development to the discussion portion of this paper to address this.
- 2) One critical experiment that requires additional information in the text is the generation of the doxycycline induced CRISPR-iPSC cell line with the GLI2 missense mutation. The authors do not discuss whether or not they generate their control line in a similar fashion to their GLI2 P>L line (such as doxycycline selection). These steps are important as they can also account for differences in endocrine differentiation.
- 3) On page 6 of the manuscript, the authors comment on unchanged proliferation and apoptosis in their final insulin-positive cell population (Fig 2F and Supp Fig 2A). It is well reported that cells differentiated to this end stage will have low proliferation, and the impact of the GLI2 P>L mutation may be more

informative at the pancreatic progenitor stage. In addition, a final cell number at the end of differentiation would be beneficial to determine whether GLI2 P>L affected cell proliferation or apoptosis at earlier stages.

4) The authors note that PDX1 and NKX6-1 expression was significantly lower in the pancreatic progenitor stage (fig 2c). Therefore, including a flow profile for PDX1 and NKX6-1 would be useful to assessing whether the generation of pancreatic progenitor is impaired. If this is the case, one would conclude that the mutation impairs pancreatic specification and as a consequence leads to fewer endocrine progenitors. Additionally did the authors consider examining non-pancreatic lineages that may instead arise from the population (such as stomach and intestines)?

5) An interesting result from these studies that the author doesn't comment on is that while endocrine fate is perturbed, their qRT-PCR analyses show up-regulation of both glucagon (Supp Fig 3C) and the transcription factor ISL1 (Supp Fig 2F), which may indicate that endocrine differentiation is not completely compromised by this mutation. Any comment on this in the discussion would be ideal.

6) What is the frequency of alpha cells in the mutant and control lines?

7) Page 7 Given that fewer beta cells are generated using the hiPSC line carrying the GLI2 P>L mutation, it is not surprising that they have a poor response to glucose challenge (Fig 2h) and I am not convinced that the authors can conclude that the mutation leads to impaired beta cell function.

8) The authors have a gene listed as GLC in Fig. 4C and Supp Fig. 3C. The authors state that GLC stands for glucagon in Supp Fig. 1G. However, GLC is not the correct gene name for glucagon and should be changed to GCG.

9) Page 9 I would suggest changing the sentence "wnt5a signaling regulation is critical for endocrine development" To "wnt5a signaling regulation is critical for pancreatic progenitor development".

Reviewer #2 (Remarks to the Author):

The manuscript by Mueller et al describes the identification of a human GLI2 allele that is associated with diabetes in a consanguineous family. Interestingly, both parents carry a heterozygous mutation, but only the father developed diabetes. The authors demonstrate that the mutation reduces the transcriptional activity of GLI2. The study goes on to show that GLI2 is expressed in a small subset of Pdx1+ progenitors in the developing human pancreatic islet and during human stem cell differentiations into beta cells. The authors then use Crispr to introduce the mutation into a hiPSC line. Phenotypic analysis demonstrates that both the heterozygous and homozygous mutant cells have impaired pancreas progenitor formation and a reduction in many of the essential pancreatic transcription factors. Unbiased transcriptome analysis identified an upregulation of the non-canonical Wnt signaling pathway and the authors confirm that inhibition of this pathway can partially rescue the GLI2 mutant endocrine progenitors. The manuscript is well written and the data is clearly presented. Data supporting the reduction in endocrine progenitor marker expression is strong and it is not surprising that this leads to later defects in beta cell differentiation. One of the strongest aspects of the study is the identification that the non-canonical Wnt pathway is upregulated in the GLI2 mutant cells and the ability to partially rescue the defect using a Wnt inhibitor. Overall, this is an interesting study that contributes to our understanding of human pancreas development; however, the normal role of GLI2 during pancreas

development needs to be clarified and there are some aspects of the study that could be improved.

Major issues:

1. The GLI2 expression data is not very convincing. The IF staining of the human tissue in Figure 1d is poor quality and although this may be due to the quality of the antibody, only a very few cells appear to be positive for GLI2. The RNA expression data in the differentiating cells (Supp Figure 1) is also relatively low compared to known transcription factor expression. The authors should mine some of the published human islet gene expression data sets to support their own expression data.
2. The nature of the GLI2 mutation is not adequately explored or discussed. The authors demonstrate that the mutation impairs the ability of GLI2 to activate transcription – which might imply it is a loss of function mutation. Since the heterozygote individuals and cells have such a robust phenotype – this could be due to haploinsufficiency. On the other hand, the MIN6 data showing that overexpression of mutant GLI2 impairs beta cell gene expression, might suggest a dominant negative effect (against other GLI proteins?). This interpretation is also supported by the fact that GLI2 expression is actually elevated in the mutant differentiated iPSCs. This would also be consistent with the apparent normally low expression of GLI2 in human cells. An optimal experiment to test this is to create an iPSC line carrying a null mutation of GLI2 – although this could be the subject of another study. At the very least, the authors should discuss how they think one copy of the mutant allele can be contributing the phenotype. An explanation for the upregulation of GLI2 expression should also be included.
3. If the authors believe that the GLI2 variant is a loss of function, the MIN6 experiments don't make a lot of sense and should be removed.
4. The nomenclature of the mutant iPSC lines is very confusing: +/- usually denotes a wildtype allele. The authors should use a more precise designation of the alleles.
5. Given that there is defective endocrine progenitor specification, it is likely that the defect is occurring prior to d14 in the earlier stage cells. Transcriptome analysis should be performed at an earlier stage to identify the primary targets of GLI2.
6. Glucagon appears to be upregulated in the mutant differentiations. IF analysis for glucagon should be performed to show the spatial expression of Glucagon. Additional IF for the other hormones would also be informative.
7. A large part of the manuscript is devoted to the beta cell differentiation phenotype. First, it isn't surprising that beta cell differentiation is impaired since there is defective endocrine progenitor specification. Second, the later stage differentiations (for both protocols) are quite poor even in the wild type cells, which calls into question the overall validity of these experiments.
8. All of the flow cytometry plots appear to be gated incorrectly. The gates should be determined using negative control cells and not be set at the center of double positive cells.

Reviewer #3 (Remarks to the Author):

In this work by Mueller et al. the authors describe the impact of a missense variant in the GLI2 gene in in vitro pancreatic development, with potential consequences for diabetes susceptibility. There is detailed

stem-cell work which shows a convincing impact of the identified variant on differentiation to pancreatic beta-like cells.

Whilst the work is of interest, I have several recommendations to improve clarity to readers of Nature Communications. Some of the data presented also require more extensive discussion. Given the potential clinical impact of the work, it is particularly important to clarify whether the authors claim or do not claim that the variant is causal and why, discussing the technical limitations of the genetic analysis.

Specific comments are provided below:

1. The abstract could benefit from a brief description of the WNT5A results, which are quite interesting. I would also recommend using consistent language to describe the phenotype, as in the abstract it's described as "paediatric hyperglycaemic patient and family members" while in the rest of the manuscript it is referred to as diabetic.
2. In the introduction (and elsewhere where applicable), the authors should use more nuanced language to describe the type of genetic variation that associates with type 2 diabetes (T2D). There is a mention to mutations of TF genes causing T2D, but this likely means that such genes are also loci that harbour common risk variants for T2D. There's no single mutation that causes T2D. Similarly, the last sentence of the introduction ("(...) identify a novel variant that predisposes to diabetes.") should be revised, as predisposition suggests a multifactorial/polygenic background (as observed in T2D) whereas this family seems to be an example of early onset diabetes of unknown cause (MODY-X). If the MODY terminology is not applicable to this family/proband, the authors should explain it.
3. A major point that requires further development is the genetics section of the paper. The rationale to home in on this GLI2 variant is not provided in enough detail. From the results and methods sections of the manuscript, it is unclear how the variant was prioritised. Have the authors fully discarded that other rare variants may be at play here? Importantly, given the family's consanguinity, have recessive variants been investigated? This mode of inheritance would be more consistent with having 2 affected siblings but not an affected parent. The family tree is also incomplete and does not provide enough information to understand the level of consanguinity in the couple. The authors should also provide further details on the age of onset for the different family members who have been diagnosed with diabetes. The possibility of other variants at play (especially considering that the mother did not develop diabetes) and the limitations of exome sequencing should be discussed in more detail by the authors. If the authors hypothesize that what is happening in this family is a case of incomplete penetrance, this point should be more extensively discussed.
4. The authors use findings from a T2D GWAS to support their claim of a potential link between GLI2 and diabetes. Whilst this is certainly possible, the variant rs12617659 is quite far from GLI2 and may regulate other genes in the region, thus it would be advisable to at least specify the distance between the variant and the promoter of GLI2 in the text).

5. The bioinformatics analyses to support that NM_001371271.1:c.4661C>T is deleterious are quite convincing, but it would be useful to contrast it with other previously described mutations in this gene, as these are briefly referred to in the Introduction. This information could be provided in panel 1A, showing the positions of the other mutations – this information may assist the authors in contrasting the phenotype of their proband with the phenotypes of other patients with GLI2 mutations.

6. In terms of defining when GLI2 is expressed during pancreas differentiation, I appreciate having a good staining on such difficult to obtain tissue is not always easy, but it would be interesting if the authors could also show scRNA-seq from human pancreas development, as these data are already in the public domain (example: Gonçalves et al. Nat Comms 2021).

7. For the work with reporter assays and overexpression of the WT and mutant forms (Figure 1e,f), the authors mention these results quite briefly, but it would be interesting to link it with the fact that the missense mutation affects the TAD region, in other words, the results are consistent with a loss-of-function mutation, which also fits the rest of the story, and the authors could highlight this more.

8. The authors carried out a detailed evaluation of the impact of GLI2 haploinsufficiency in pancreatic in vitro differentiation, obtaining compelling data that links GLI2 loss with decreased expression of key developmental genes. The data presented seems very solid and just have a few of suggestions to add clarity to readers (particularly those from outside the stem cell field):

- As the authors explain, genetic background can affect differentiation efficiency and the overall phenotype observed. Can the authors provide details on the specific iPSC line that was used in the experiment?
- The authors should define the n shown in the experiments. For instance, for the differentiation data, does n=3 represent different clones, different differentiation rounds, or both?
- It is unclear what the negative control clones (GLI2CTRL) represent, were they also derived from the Cas9 line, and where they also transfected with the guide RNAs & oligos to do the knock-in of the variant?
- It would be good to show a WT clone for comparison in Supplementary Figure 1d

9. I quite welcome the validation of the results using a different differentiation protocol. But a question that perhaps remains unanswered with the current data, is whether the effects observed at the beta-like cell stage are reflective of a role of GLI2 at that stage or whether they reflect the impairment to form an appropriate pool of pancreatic progenitors. In other words, if the pool of endocrine progenitors expressing PDX1/NKX6.1 is reduced doesn't that have a domino effect on the subsequent stages, leading to less NGN3+ and then less INS+ cells? This limitation should be discussed by the authors.

10. The RNA-seq analysis presentation and description of the methodology would benefit from some improvement, as some of the panels presented in the manuscript as not very informative. Specific points:

- Full DESeq2 results should be provided in a Supplementary Table.
- Please provide the version of the reference genome used as there are several versions/patches within GRCh38.
- Gene Network Analysis: what was the p-value/FDR threshold?

- For instance, Supplementary Figure 4a does not inform on the gene, or the degree of FC/significance, a volcano plot is a more standard representation of this type of dataset overview. If representing differentially expressed genes as a heatmap, the type of clustering analysis applied should be described. It also suggests the authors carried out differential expression analysis with n=2/group, which is unusual and should be justified.
- I suggest moving the volcano plot from Figure 3a into a Supplementary Figure, since there are so few DEGs, swapping it for the more interesting data for the endocrine progenitor stage.
- I also suggest comparing the DEGs between the 2 differentiation stages, as it seems that there may be some interesting communalities (e.g., GATA6).
- The description of the gene ontology/GSEA analyses is quite brief and perhaps the order in which they are presented surprising (usually one would start with a general GO analysis then moving on to specific gene sets for GSEA). More in depth discussion of these results would be welcomed.
- There are no methods for the GSEA. What tool was used and using which parameters? Which gene sets were used? Was a database of gene sets queried, or were the gene sets pre-defined?
- Figure 3f shows a series of genes validated by qPCR, however there is no indication of the direction of differential expression in the RNA-seq, it should also be indicated for this panel whether the validations were done with different clones/differentiation rounds.
- The GSEA analysis shows very interesting results, but the authors have not described how the GSEA was carried out, what was the size of the gene set tested? Did the authors restrict the analysis to only genes expressed in that developmental stage?
- An oversight perhaps, Figures 4 and Supplementary 4 seem to be the same data with different legends, this should be revised.

11. In the final part, the authors explain that GLI2+/- endocrine progenitors showed upregulation of non-canonical WNT ligands, such as WNT5A, WNT5B, WNT7A, and WNT receptors (FZD3, FZD7, FZD8). From this result, the authors focus specifically on WNT5A, but they do not offer a reason for this. Unfortunately, I could not review the data pertaining to WNT5A modulation as the figure is missing.

Minor comments:

1. When first describing the variant, I suggest including a mention to transcript isoform, as the position of the variant can change. I would say “NM_001371271.1:c.4661C>T” (see other isoforms here showing that the position does change: https://www.ncbi.nlm.nih.gov/snp/rs767802807#variant_details). The authors should also mention that this is a rare variant and provide its MAF in the text rather than just in a figure legend.
2. Legend of figure 1: “The heterozygous GLI2 p.P1554L (c.C4661T) variant was found in four individuals (black symbols) of a consanguineous family with incomplete penetrance.” Is likely incorrect as the grandmother was not genotyped.
3. Supplementary Figure 1: the logic of using different colours for the bars represented could be clearer by adding a legend on the side, or least describe it in the legend. The abbreviations for the different stages should also be defined in the legend (e.g., GT = Gut tube).
4. Discussion: sub-set should be subset
5. The reference to the deceased grandmother could perhaps be improved from “Moreover, the grandmother paternal side had diabetes, but passed away and, therefore, was not available for

sequencing” to something more like “Moreover, the grandmother from the paternal side also had [SPECIFY TYPE] diabetes. However, we were not able to determine her genotype as she was deceased.”

Inês Cebola

Point-by-point response to the reviewers' comments

We thank the Reviewers for their thorough evaluation of our manuscript and insightful comments, addressing them has substantially improved our study. We have carried out new experiments and revised the text to address all the concerns raised. These have resulted in multiple new figure panels, 2 new supplementary figures and 3 new supplementary tables. Below is a point-by-point response to the reviewers' concerns with our responses shown in blue. We hope the reviewers and editor will find the revised manuscript now suitable for publication in *Nature Communications*.

REVIEWER COMMENTS

Reviewer #1 (Remarks to the Author):

The manuscript submitted by Mueller et al. presents a novel analysis of a missense mutation in the Hedgehog signalling transcription factor GLI2 that was first observed in clinical diagnosis of diabetes with an unknown cause. The authors link the presence of GLI2-associated activity to endocrine and beta cell commitment using a GLI2-mutated human induced pluripotent stem cell line with CRISPR-editing. To investigate downstream mechanisms underlying the impairment in pancreatic differentiation the authors perform RNAsequencing at various stages of differentiation and uncover differences in the expression levels of non-canonical wnt ligands and further demonstrate that inhibition of non-canonical wnt could rescue pancreatic differentiation in the iPSC line carrying the missense mutation. The data are well presented. I am unsure however about the developmental stage impacted by the mutation. The authors suggest that GLI2 mutation impairs endocrine progenitor development, I would argue that the impairment occurs earlier during the formation of the pancreatic progenitor cells, as the authors detect lower NKX6-1 expression already between d9-d11 of differentiation (Fig 2c and suppl fig 3b). I would expect a lower frequency of PDX1+/NKX6-1+ cells already at the pancreatic progenitor stage which would manifest in fewer endocrine progenitor cells at day 14. Their rescue experiment is consistent with this idea, as they modulate non-canonical wnt5a prior to pancreatic progenitor differentiation rather than during endocrine progenitor specification.

We would like to thank the reviewer for the very constructive suggestions that helped us to improve our manuscript. We appreciate all positive comments and answer below to each one of the points raised.

Suggestions to strengthen the manuscript:

1) A major question that is not clear throughout the text is the role of GLI2 in Hedgehog signalling and how the Hedgehog signalling axis is affected with the GLI2 P>L mutation. The authors do show a GLI responsive luciferase assay, but they fail to comment on possible compensation or modulation through GLI1 or GLI3 in GLI2 P>L cells. There is interesting information provided in the RNA sequencing data provided in Fig. 3 and Supp. Fig. 4 that suggests down-regulation of GLI1 and up-regulation of GLI2 and GLI3. The authors do comment on page 5 of the manuscript regarding the transcriptional activator or repressor roles of GLI, but any further characterization of what is happening to Hedgehog signalling in GLI2 P>L cells is lacking. Is SANT-1 required during the differentiation of the mutant lines? What happens in the absence of SANT-1? The authors could also benefit from adding more information regarding

Hedgehog signalling in pancreatic and islet development to the discussion portion of this paper to address this.

We thank the reviewer for raising this important point. In the revised study, we expanded the analysis of the consequences of GLI2 P>L mutation on the Hedgehog signalling pathway. First, we performed the luciferase reporter assay in the iPSC line carrying the mutant variant in addition to the HEK293T cells. Similarly, we found that GLI2^{P>LHET} iPSCs displayed reduced endogenous transcriptional activation of the GLI-responsive luciferase reporter compared to control iPSCs. The data are now included in the new Supplementary Fig. 2a.

Second, we further characterized the dysregulation of the GLI transcripts in GLI2^{P>LHET} iPSCs undergoing differentiation as a possible compensation mechanism to the HH downstream inactivation. Specifically, we found that the upregulation of GLI2 starts at endocrine progenitor stage (see new Supplementary Fig. 2i) and showed all DEGs related to the Hedgehog components in new Supplementary Table 6.

Third, in the revised Results section, we included more details about the differentiation protocols used throughout the study and in which experiment the antagonist of Hedgehog pathway, SANT-1, was used or not. Even if most of our study is based on the Russ et al. 2015 protocol (in the absence of SANT-1), we found that the Reznia et al. 2014 protocol (in the presence of SANT-1) leads into a similar compromised pancreatic and β -cells development of the mutant iPSCs.

Finally, we expanded the discussion about the HH signaling in pancreas/islet development and the overall negative impact of GLI2^{P>L} variant on the HH signaling, which leads to a deregulation of the fine balance between the GLI factors (see page 11 of the new manuscript).

2) One critical experiment that requires additional information in the text is the generation of the doxycycline induced CRISPR-iPSC cell line with the GLI2 missense mutation. The authors do not discuss whether or not they generate their control line in a similar fashion to their GLI2 P>L line (such as doxycycline selection). These steps are important as they can also account for differences in endocrine differentiation.

We agree with the reviewer. This is a very important information and we apologize that it was not properly explained in the first version of the manuscript.

The control iPSC line (HMGUi001-A2) has the same background as GLI2^{P>L} mutant line. This line carries a doxycycline-inducible Cas9 expression system integrated in the AAVS1 locus. The information is now included in the Methods section of the revised manuscript.

Moreover, we exposed the control iPSC line to dox-selection and compared its ability to differentiate *versus* untreated control cells. This important experiment showed no impact of the dox-selection on the differentiation protocol (see new Supplementary Fig. 1g).

3) On page 6 of the manuscript, the authors comment on unchanged proliferation and apoptosis in their final insulin-positive cell population (Fig 2F and Supp Fig 2A). It is well reported that cells differentiated to this end stage will have low proliferation, and the impact of the GLI2 P>L mutation may be more informative at the pancreatic progenitor stage. In addition, a final cell number at the end of differentiation would be beneficial to determine whether GLI2 P>L affected cell proliferation or apoptosis at earlier stages.

In compliance with the reviewer's request, we expanded the analysis of proliferation and apoptosis in $GLI2^{P>L}$ mutant cells at pancreatic progenitor stage. No significant changes in proliferation were measured in the $GLI2^{P>L}$ mutant cells compared to controls at both pancreatic and endocrine progenitor stages, while the number of apoptotic cells was significantly increased at pancreatic progenitor stage. The new results have been included in new Supplementary Fig. 2d-g.

We did not assess the final cell number at the end of differentiation, as we typically collect cell clusters throughout the experiment to perform intermediate time-points analysis by RT-qPCR or IF. Thus, we are afraid the final cell number might be inaccurate and not representative.

4) The authors note that PDX1 and NKX6-1 expression was significantly lower in the pancreatic progenitor stage (fig 2c). Therefore, including a flow profile for PDX1 and NKX6-1 would be useful to assessing whether the generation of pancreatic progenitor is impaired. If this is the case, one would conclude that the mutation impairs pancreatic specification and as a consequence leads to fewer endocrine progenitors. Additionally did the authors consider examining non-pancreatic lineages that may instead arise from the population (such as stomach and intestines)?

We agree with the reviewer that cell differentiation is impaired in $GLI2^{P>L}$ mutant cells already at pancreatic progenitor stage, which in turn affects the number of endocrine progenitors. This has been now better explained in the revised manuscript (see new Results and Discussion sections). Unfortunately, while *NKX6-1* transcript levels can be measured at day 9 of differentiation (Fig. 2, Supplementary Fig.4, 5), we have difficulties in detecting the protein by IF or Flow-cytometry at the same stage. In our hands the antibody works well starting from d12-14, the Endocrine progenitor stage. In compliance with the reviewer's request, we examined other endoderm-derivative lineages by RT-qPCR and found no induction of liver, stomach, and intestine markers in $GLI2^{P>L}$ mutant cells (see new Supplementary Fig. 2j). This is in line with the RNASeq analysis (see the new Supplementary Tables 5 and 6).

5) An interesting result from these studies that the author doesn't comment on is that while endocrine fate is perturbed, their qRT-PCR analyses show up-regulation of both glucagon (Supp Fig 3C) and the transcription factor ISL1 (Supp Fig 2F), which may indicate that endocrine differentiation is not completely compromised by this mutation. Any comment on this in the discussion would be ideal.

This is a very interesting remark; the difference in ISL1 or Glucagon gene expression in mutant cells might be linked to the genotype ($GLI2^{P>L}$ HET vs HOMO) and differentiation protocols, respectively.

In new Fig. 2e, g, we included the RT-qPCR of *ISL1* and Glucagon in $GLI2^{P>L}$ HET and added one sentence about the difference in modulation of *ISL1* between $GLI2^{P>L}$ HET and HOMO in new Supplementary Fig. 3. Also, we explained the difference in Glucagon expression according to the differentiation protocols used (see page 8 of the Results section of the revised manuscript).

6) What is the frequency of alpha cells in the mutant and control lines?

In compliance with the reviewer's request, we measured the frequency of alpha cells in GLI2^{P>L} and control cells. This is now included in new Supplementary Fig. 2h.

7) Page 7 Given that fewer beta cells are generated using the hiPSC line carrying the GLI2 P>L mutation, it is not surprising that they have a poor response to glucose challenge (Fig 2h) and I am not convinced that the authors can conclude that the mutation leads to impaired beta cell function.

The reviewer is correct, we revised the text of the Results section accordingly to his/her comment (see page 7 of the revised manuscript).

8) The authors have a gene listed as GLC in Fig. 4C and Supp Fig. 3C. The authors state that GLC stands for glucagon in Supp Fig. 1G. However, GLC is not the correct gene name for glucagon and should be changed to GCG.

We apologize for the mistake; Glucagon abbreviation has been edited in the text and figures of the revised manuscript.

9) Page 9 I would suggest changing the sentence "wnt5a signaling regulation is critical for endocrine development" To "wnt5a signaling regulation is critical for pancreatic progenitor development".

The sentence has been edited in compliance with the reviewer's suggestion.

Reviewer #2 (Remarks to the Author):

The manuscript by Mueller et al describes the identification of a human GLI2 allele that is associated with diabetes in a consanguineous family. Interestingly, both parents carry a heterozygous mutation, but only the father developed diabetes. The authors demonstrate that the mutation reduces the transcriptional activity of GLI2. The study goes on to show that GLI2 is expressed in a small subset of Pdx1+ progenitors in the developing human pancreatic islet and during human stem cell differentiations into beta cells. The authors then use Crispr to introduce the mutation into a hiPSC line. Phenotypic analysis demonstrates that both the heterozygous and homozygous mutant cells have impaired pancreas progenitor formation and a reduction in many of the essential pancreatic transcription factors. Unbiased transcriptome analysis identified an upregulation of the non-canonical Wnt signaling pathway and the authors confirm that inhibition of this pathway can partially rescue the GLI2 mutant endocrine progenitors. The manuscript is well written and the data is clearly presented. Data supporting the reduction in endocrine progenitor marker expression is strong and it is not surprising that this leads to later defects in beta cell differentiation. One of the strongest aspects of the study is the identification that the non-canonical Wnt pathway is upregulated in the GLI2 mutant cells and the ability to partially rescue the defect using a Wnt inhibitor. Overall, this is an interesting study that contributes to our understanding of human pancreas development; however, the normal role of GLI2 during pancreas development needs to be clarified and there are some aspects of the study that could be improved.

We are grateful to the reviewer for all his/her constructive suggestions that helped us to further improve our manuscript. We answer below to each one of the points raised.

Major issues:

1. The GLI2 expression data is not very convincing. The IF staining of the human tissue in Figure 1d is poor quality and although this may be due to the quality of the antibody, only a very few cells appear to be positive for GLI2. The RNA expression data in the differentiating cells (Supp Figure 1) is also relatively low compared to known transcription factor expression. The authors should mine some of the published human islet gene expression data sets to support their own expression data.

We thank the reviewer for suggesting this very important experiment. We expanded the spatiotemporal analysis of GLI2 in human fetal pancreatic tissue by IF staining. The IF has been now performed at later developmental stages, 12pcw and 20pcw, and with an additional antibody against GLI2. Both anti-GLI2 antibodies show the same results, with an increasing number of cells being positive for GLI2 at later time points. The new results are shown in new Fig. 1e.

Due to the low RNA capture rate in some scRNA-seq technologies, generally TFs with low levels of expression may not be detected reliably (Hicks et al 2018; Shrestha et al JCI 2021). Regarding *GLI2* transcript level, we agree with the reviewer its levels of expression seem to be low in both RT-qPCR and scRNA-seq.

2. The nature of the GLI2 mutation is not adequately explored or discussed. The authors demonstrate that the mutation impairs the ability of GLI2 to activate transcription – which might imply it is a loss of function mutation. Since the heterozygote individuals and cells have such a robust phenotype – this could be due to haploinsufficiency.

On the other hand, the MIN6 data showing that overexpression of mutant GLI2 impairs beta cell gene expression, might suggest a dominant negative effect (against other GLI proteins?). This interpretation is also supported by the fact that GLI2 expression is actually elevated in the mutant differentiated iPSCs. This would also be consistent with the apparent normally low expression of GLI2 in human cells. An optimal experiment to test this is to create an iPSC line carrying a null mutation of GLI2 – although this could be the subject of another study. At the very least, the authors should discuss how they think one copy of the mutant allele can be contributing the phenotype. An explanation for the upregulation of GLI2 expression should also be included.

We thank the reviewer for raising these very important points. In the revised manuscript, we have expanded the Discussion about the consequences of GLI2 P>L mutation on the Hedgehog signalling pathway, haploinsufficiency as underlying mechanism of disease and incomplete penetrance (see pages 10-11 of the revised manuscript).

In addition, we performed additional experiments to assess the effects of the GLI2 P>L mutation. First, we carried out a luciferase reporter assay in the iPSC line carrying the mutant variant. Like in HEK293T cells, we found that the GLI2^{P>LHET} iPSCs display reduced endogenous transcriptional activation of the GLI-responsive luciferase reporter in comparison to control iPSCs. The data are included in the new Supplementary Fig. 2a.

Second, we characterized more extensively the dysregulation of the GLI transcripts in GLI2^{P>LHET} iPSCs undergoing differentiation. New Supplementary Fig. 2i shows that the upregulation of *GLI2* starts only late, at endocrine progenitor stage. New

Supplementary Table 6 includes all DEGs related to the Hedgehog components. Finally, we discuss about the upregulation of *GLI2* expression as a possible compensation of HH downstream inactivation. Taken together, our findings indicate an overall negative impact of *GLI2*^{P>L} variant on the HH signaling, which is accompanied by a deregulation of the fine balance between the GLI factors.

Regarding the iPSC line carrying a null mutation of *GLI2*, we agree with the reviewer that this would have been an ideal experiment to fully assess the nature of the mutation. However, our attempts to generate a CRISPR/Cas9-mediated knockout of *GLI2* gene in iPSCs were unsuccessful; so far, no viable KO clone was established. This is actually the main reason of our delay in resubmission.

3. If the authors believe that the *GLI2* variant is a loss of function, the MIN6 experiments don't make a lot of sense and should be removed.

The reviewer is correct; the MIN6 experiments have been removed from the manuscript.

4. The nomenclature of the mutant iPSC lines is very confusing: +/+ usually denotes a wildtype allele. The authors should use a more precise designation of the alleles.

We agree with the reviewer. In compliance with his/her suggestion, we modified the nomenclature the mutant iPSC lines throughout the revised text and figures and refer to them as *GLI2*^{P>L HET} and *GLI2*^{P>L HOMO}.

5. Given that there is defective endocrine progenitor specification, it is likely that the defect is occurring prior to d14 in the earlier stage cells. Transcriptome analysis should be performed at an earlier stage to identify the primary targets of *GLI2*.

We agree with the reviewer that *GLI2*^{P>L} mutant cells start having defects in cell differentiation from the pancreatic progenitor stage onward. This has been now better explained in the revised manuscript (see new Results and Discussion sections). In the original study, we decided to perform the transcriptome analysis at early stage (Gut tube stage) and at endocrine progenitor stage to 1) unveil the initial events that might underlie the phenotype and 2) better characterize the mutant lines, respectively. In the revised manuscript, we included the few DEG targets in common between the two stages, such as *GATA6* and *WNT5a* (see new Supplementary Tables 5 and 6). Moreover, we performed RT-qPCR to validate these common DEG signatures and dysregulated WNT and HH pathway components at pancreatic progenitor stage. The new data has been included in the new Supplementary Fig. 5e.

6. Glucagon appears to be upregulated in the mutant differentiations. IF analysis for glucagon should be performed to show the spatial expression of Glucagon. Additional IF for the other hormones would also be informative.

In compliance with the reviewer's request, we included IF staining for Glucagon and measured the frequency of alpha cells in *GLI2*^{P>L} and control cells. This is included in new Supplementary Fig. 2.

The reviewer is right. Glucagon expression is unchanged or slightly induced in *GLI2*^{P>L} when the Rezania differentiation protocol was used (Supplementary Fig.4). This

discrepancy has been now better explained in the Results section of the revised manuscript (see page 8).

7. A large part of the manuscript is devoted to the beta cell differentiation phenotype. First, it isn't surprising that beta cell differentiation is impaired since there is defective endocrine progenitor specification. Second, the later stage differentiations (for both protocols) are quite poor even in the wild type cells, which calls into question the overall validity of these experiments.

We agree with the reviewer that the impairment in pancreatic cell differentiation occurs in $GLI2^{P>L}$ mutant cells at the pancreatic progenitor stage. This in turn affects the number of endocrine progenitors. In the revised manuscript (see new Results and Discussion sections), we shifted the focus to the $GLI2^{P>L}$ variant consequences at pancreatic progenitor stage rather than beta-cell function.

8. All of the flow cytometry plots appear to be gated incorrectly. The gates should be determined using negative control cells and not be set at the center of double positive cells.

We apologize with the reviewer for the lack of clarity in our manuscript. The gates were always determined using negative control cells. For gating, unstained samples were used as negative controls. Representative flow cytometry pseudocolor plots and gating strategy have been now included in Supplementary Fig. 6 of the revised manuscript.

Reviewer #3 (Remarks to the Author):

In this work by Mueller et al. the authors describe the impact of a missense variant in the $GLI2$ gene in in vitro pancreatic development, with potential consequences for diabetes susceptibility. There is detailed stem-cell work which shows a convincing impact of the identified variant on differentiation to pancreatic beta-like cells. Whilst the work is of interest, I have several recommendations to improve clarity to readers of Nature Communications. Some of the data presented also require more extensive discussion. Given the potential clinical impact of the work, it is particularly important to clarify whether the authors claim or do not claim that the variant is causal and why, discussing the technical limitations of the genetic analysis.

We thank the reviewer for recognizing the value of our study and for all the valuable suggestions which have helped us to improve our manuscript.

Specific comments are provided below:

1. The abstract could benefit from a brief description of the $WNT5A$ results, which are quite interesting. I would also recommend using consistent language to describe the phenotype, as in the abstract it's described as "paediatric hyperglycaemic patient and family members" while in the rest of the manuscript it is referred to as diabetic.

We thank the reviewer for both suggestions. We revised the abstract to include the $WNT5A$ results. Additionally, in the revision we described the clinical phenotype in a

more consistent manner. Specifically, we refer to the phenotype as 'early-onset and insulin-dependent diabetes of unknown cause'.

2. In the introduction (and elsewhere where applicable), the authors should use more nuanced language to describe the type of genetic variation that associates with type 2 diabetes (T2D). There is a mention to mutations of TF genes causing T2D, but this likely means that such genes are also loci that harbour common risk variants for T2D. There's no single mutation that causes T2D. Similarly, the last sentence of the introduction (“(...) identify a novel variant that predisposes to diabetes.”) should be revised, as predisposition suggests a multifactorial/polygenic background (as observed in T2D) whereas this family seems to be an example of early onset diabetes of unknown cause (MODY-X). If the MODY terminology is not applicable to this family/proband, the authors should explain it.

We understand the concerns of the reviewer and agree with the importance of employing the correct terminology. In the revision, we edited the 'Introduction' section and specified that 'genetic variants in or nearby some pancreatic transcription factor genes have been associated with risk for type 2 diabetes' instead of 'causing type 2 diabetes'. However, we believe that the definition “diabetes of unknown origin” better applies to this family harbouring the GLI2P>L instead of the MODY terminology, which is usually inherited in an autosomal dominant fashion and patients have heterozygous mutations.

3. A major point that requires further development is the genetics section of the paper. The rationale to home in on this GLI2 variant is not provided in enough detail. From the results and methods sections of the manuscript, it is unclear how the variant was prioritised. Have the authors fully discarded that other rare variants may be at play here? Importantly, given the family's consanguinity, have recessive variants been investigated? This mode of inheritance would be more consistent with having 2 affected siblings but not an affected parent. The family tree is also incomplete and does not provide enough information to understand the level of consanguinity in the couple. The authors should also provide further details on the age of onset for the different family members who have been diagnosed with diabetes. The possibility of other variants at play (especially considering that the mother did not develop diabetes) and the limitations of exome sequencing should be discussed in more detail by the authors. If the authors hypothesize that what is happening in this family is a case of incomplete penetrance, this point should be more extensively discussed.

We agree with the reviewer about the limitations of our genetics analysis. In compliance with his/her request, we have included the age of onset of diabetes, where available. Also, we clarified that no mutation in known, diabetes-causing genes have been found in any of the four family members (father, mother, and siblings) (see Results section and Fig. 1 legend of the revised manuscript). Unfortunately, we have no information about the level of consanguinity in the couple, as the clinicians, co-authors in the study, have difficulties to access the family's information and to re-establish any contact with them. Finally, we followed the reviewer's suggestion and expanded the Discussion section including some remarks about haploinsufficiency as underlying mechanism of the disease and incomplete penetrance (see pages 10-11 of the new manuscript).

4. The authors use findings from a T2D GWAS to support their claim of a potential link between *GLI2* and diabetes. Whilst this is certainly possible, the variant rs12617659 is quite far from *GLI2* and may regulate other genes in the region, thus it would be advisable to at least specify the distance between the variant and the promoter of *GLI2* in the text).

The reviewer is correct. The variant rs12617659 is far from *GLI2* (at approximately 200.000Kb distance) and any direct gene regulation would be at this point just speculation. We therefore decided to remove the sentence in the revised manuscript.

5. The bioinformatics analyses to support that NM_001371271.1:c.4661C>T is deleterious are quite convincing, but it would be useful to contrast it with other previously described mutations in this gene, as these are briefly referred to in the Introduction. This information could be provided in panel 1A, showing the positions of the other mutations – this information may assist the authors in contrasting the phenotype of their proband with the phenotypes of other patients with *GLI2* mutations.

We thank the reviewer for raising this point. However, after an initial attempt, we have respectfully decided not to include the previously described mutations in *GLI2* gene in the cartoon shown in Fig. 1A. We had difficulties in showing all the *GLI2* variants reported and found that this additional information might eventually create some confusion in the visualization of the *GLI2* c.4661C>T mutation reported here. However, we included the link to OMIM 11 selected mutations associated with Holoprosencephaly and Culler-Jones syndrome (<https://omim.org/allelicVariants/165230>) in the revised Supplementary Table 1.

6. In terms of defining when *GLI2* is expressed during pancreas differentiation, I appreciate having a good staining on such difficult to obtain tissue is not always easy, but it would be interesting if the authors could also show scRNA-seq from human pancreas development, as these data are already in the public domain (example: Gonçalves et al. Nat Comms 2021).

In compliance with the reviewer's request, in the revision we expanded the IF analysis of *GLI2* in human fetal pancreatic tissue to later developmental stages, 12pcw and 20pcw, and confirmed the protein distribution with another antibody against *GLI2*. Both anti-*GLI2* antibodies showed the same results, with more cells being positive for *GLI2* at later time points. The results are now shown in new Fig. 1e.

Due to the low RNA capture rate by some scRNA-seq technologies, TFs with low levels of expression may not be detected reliably (Hicks et al 2018; Shrestha et al JCI 2021). Accordingly, we found the *GLI2* transcript levels of expression to be quite low in scRNA-seq datasets.

7. For the work with reporter assays and overexpression of the WT and mutant forms (Figure 1e,f), the authors mention these results quite briefly, but it would be interesting to link it with the fact that the missense mutation affects the TAD region, in other words, the results are consistent with a loss-of-function mutation, which also fits the rest of the story, and the authors could highlight this more.

We thank the reviewer for the valuable suggestion. In the revision, we performed additional experiments to assess the effects of the *GLI2* P>L mutation on Hedgehog

transcriptional activity. We carried out a luciferase reporter assay in the iPSC line carrying the mutant variant. Like in HEK293T cells, we found that the GLI2^{P>L} iPSCs display reduced endogenous transcriptional activation of the GLI-responsive luciferase reporter in comparison to control iPSCs. The data are included in the new Supplementary Fig. 2a. Additionally, we highlighted the consequences of GLI2 P>L mutation in the TAD domain on the Hedgehog signalling pathway in the revised Discussion section (see page 11).

8. The authors carried out a detailed evaluation of the impact of GLI2 haploinsufficiency in pancreatic in vitro differentiation, obtaining compelling data that links GLI2 loss with decreased expression of key developmental genes. The data presented seems very solid and just have a few of suggestions to add clarity to readers (particularly those from outside the stem cell field):

- As the authors explain, genetic background can affect differentiation efficiency and the overall phenotype observed. Can the authors provide details on the specific iPSC line that was used in the experiment?

We agree with the reviewer. This is a very important information and we apologize that it was not properly explained in the first version of the manuscript. The information is now included in the Methods section of the revised manuscript. Briefly, the control iPSC line (HMGUi001-A2) has the same background as GLI2^{P>L} mutant line. This line carries a doxycycline inducible Cas9 expression system integrated in the AAVS1 locus.

- The authors should define the n shown in the experiments. For instance, for the differentiation data, does n=3 represent different clones, different differentiation rounds, or both?

This is also very important point; the information has been now included in the Figure legends of the revised manuscript and Methods section.

Two heterozygous and two homozygous mutant HMGUi001-A2 iPSC lines carrying the c.C4661T point mutation in GLI2 were established and compared to the isogenic control iPSC line for their pluripotency and differentiation properties. Differentiation experiments were carried out at least 3 times on both heterozygous and homozygous independent clones.

- It is unclear what the negative control clones (GLI2CTRL) represent, were they also derived from the Cas9 line, and where they also transfected with the guide RNAs & oligos to do the knock-in of the variant?

Also see answers above. As additional control, we exposed the control iPSC line to dox-selection and compared its ability to differentiate *versus* untreated control cells. This control experiment showed no impact of the dox-selection on the differentiation protocol. This important experiment is now shown in new Supplementary Fig. 1g.

- It would be good to show a WT clone for comparison in Supplementary Figure 1d. This has been now added to the revised Supplementary Figure 1d.

9. I quite welcome the validation of the results using a different differentiation protocol. But a question that perhaps remains unanswered with the current data, is whether the effects observed at the beta-like cell stage are reflective of a role of GLI2 at that stage or whether they reflect the impairment to form an appropriate pool of pancreatic progenitors. In other words, if the pool of endocrine progenitors expressing

PDX1/NKX6.1 is reduced doesn't that have a domino effect on the subsequent stages, leading to less NGN3+ and then less INS+ cells? This limitation should be discussed by the authors.

The reviewer is correct, the impairment in pancreatic cell differentiation occurs in GLI2^{P>L} mutant cells at early stage on the pancreatic progenitor pool, which then in turn affects the number of endocrine progenitors. In the revised manuscript (see new Results and Discussion sections), we have shifted the focus to GLI2^{P>L} variant consequences at pancreatic progenitor stage. We conclude that a diminished expression of PDX1 and NKX6.1 in patient-like iPSCs from pancreatic progenitor stage onwards is responsible for the endocrine progenitor pool depletion.

10. The RNA-seq analysis presentation and description of the methodology would benefit from some improvement, as some of the panels presented in the manuscript as not very informative. Specific points:

We thank the reviewer for the suggestions, all points have been addressed in the revision – see below.

- Full DESeq2 results should be provided in a Supplementary Table.

The results have been now provided in new Supplementary Tables 4, 5, 6.

- Please provide the version of the reference genome used as there are several versions/patches within GRCh38.

The information has been added to the revised Methods section.

- Gene Network Analysis: what was the p-value/FDR threshold?

The information has been added to the revised legend of Figure 3.

- For instance, Supplementary Figure 4a does not inform on the gene, or the degree of FC/significance, a volcano plot is a more standard representation of this type of dataset overview. If representing differentially expressed genes as a heatmap, the type of clustering analysis applied should be described. It also suggests the authors carried out differential expression analysis with n=2/group, which is unusual and should be justified.

We followed the suggestion of the reviewer and removed the heatmap shown in the old Supplementary Figure 4a and replaced it with a Volcano plot (also see answer to point below).

- I suggest moving the volcano plot from Figure 3a into a Supplementary Figure, since there are so few DEGs, swapping it for the more interesting data for the endocrine progenitor stage.

In compliance with the reviewer's request, we generated a Volcano plot for the DEGs at endocrine progenitor stage (now in Fig. 3b) and swapped it with the one showing gut tube stage DEGs (now in Supplementary Fig. 5a) (also see answer to point above).

- I also suggest comparing the DEGs between the 2 differentiation stages, as it seems that there may be some interesting communalities (e.g., GATA6).

We compared common DEGs between the two differentiation stages and we found indeed some interesting genes modulated in the same direction. Common DEGs,

including GATA6 and WNT5A, are now shown in the two Volcano plots (Fig. 3b and Supplementary Fig. 5a) and marked in Red in the Supplementary Tables 5 and 6.

- The description of the gene ontology/GSEA analyses is quite brief and perhaps the order in which they are presented surprising (usually one would start with a general GO analysis then moving on to specific gene sets for GSEA). More in depth discussion of these results would be welcomed.

We apologize with the reviewer about this. We have now reorganized the text describing the two analyses (GO analysis and GSEA) and their order in the revised manuscript.

- There are no methods for the GSEA. What tool was used and using which parameters? Which gene sets were used? Was a database of gene sets queried, or were the gene sets pre-defined?

We included the information about the GSEA analysis and database used in the revised Methods section.

- Figure 3f shows a series of genes validated by qPCR, however there is no indication of the direction of differential expression in the RNA-seq, it should also be indicated for this panel whether the validations were done with different clones/differentiation rounds.

We included the information in the revised legend of Figure 3. All tested genes showed concordant differential expression with the RNASeq results and were validated on an independent differentiation experiment.

- The GSEA analysis shows very interesting results, but the authors have not described how the GSEA was carried out, what was the size of the gene set tested? Did the authors restrict the analysis to only genes expressed in that developmental stage?

We included the information about the GSEA analysis and database used in the revised Methods section. The analysis was restricted to the EP stage.

- An oversight perhaps, Figures 4 and Supplementary 4 seem to be the same data with different legends, this should be revised.

We apologize with the reviewer for this terrible mistake. It was an oversight, we uploaded twice the same figure (Figure 3). We fixed the issue as soon as we realised about it and resubmitted to the journal the correct Figure 4, but I assume this was too late and the manuscript already sent to this reviewer.

11. In the final part, the authors explain that GLI2[±] endocrine progenitors showed upregulation of non-canonical WNT ligands, such as WNT5A, WNT5B, WNT7A, and WNT receptors (FZD3, FZD7, FZD8). From this result, the authors focus specifically on WNT5A, but they do not offer a reason for this. Unfortunately, I could not review the data pertaining to WNT5A modulation as the figure is missing.

The rationale behind focusing on WNT5A was: 1) it was the most upregulated non-canonical Wnt genes in GLI2^{P>L} mutant line 2) at both temporal stages; and 3) previous literature suggesting its role in endocrine development.

Minor comments:

1. When first describing the variant, I suggest including a mention to transcript isoform, as the position of the variant can change. I would say “NM_001371271.1:c.4661C>T” (see other isoforms here showing that the position does change: https://www.ncbi.nlm.nih.gov/snp/rs767802807#variant_details). The authors should also mention that this is a rare variant and provide its MAF in the text rather than just in a figure legend.

The information was inserted in revised manuscript (see Results section pages 4-5).

2. Legend of figure 1: “The heterozygous GLI2 p.P1554L (c.C4661T) variant was found in four individuals (black symbols) of a consanguineous family with incomplete penetrance.” Is likely incorrect as the grandmother was not genotyped.

We thank the reviewer for raising this point. Four members of the family (father, mother and two siblings) were sequenced and are indicated with a red outlined box, while black shading indicates diabetes. This is now better explained in the revised legend of Figure 1.

3. Supplementary Figure 1: the logic of using different colours for the bars represented could be clearer by adding a legend on the side, or least describe it in the legend. The abbreviations for the different stages should also be defined in the legend (e.g., GT = Gut tube).

The legend has been now inserted in the revised Supplementary Figure 1 and the abbreviation defined in the accompanying figure legend.

4. Discussion: sub-set should be subset

This has been revised in the manuscript text.

5. The reference to the deceased grandmother could perhaps be improved from “Moreover, the grandmother paternal side had diabetes, but passed away and, therefore, was not available for sequencing” to something more like “Moreover, the grandmother from the paternal side also had [SPECIFY TYPE] diabetes. However, we were not able to determine her genotype as she was deceased.”

The sentence has been revised in compliance with the reviewer’s suggestion. Unfortunately, we were not able to acquire this information from the family.

REVIEWERS' COMMENTS

Reviewer #1 (Remarks to the Author):

I thank the authors for addressing the questions raised. Overall, the authors provided data to support their responses when possible. The additional text providing a brief overview of hedgehog signalling in the pancreas and validating the effects of doxycycline exposure on differentiation in the control line is appreciated. The explanation for the absence of final cell number is understandable. Clarification of phenotypic differences and final endocrine cell development between the heterozygous and homozygous lines is presented.

Reviewer #2 (Remarks to the Author):

The authors have done an excellent job of addressing my concerns and comments. I have no further concerns.

Reviewer #3 (Remarks to the Author):

I thank the authors for addressing thoroughly most of my comments and for making substantial improvements to the manuscript during the revision. I recommend this manuscript for publication.

There are still a couple of minor points, which I believe could be better addressed:

1. In terms of the listed limitations of the study, the authors fall short of providing a comprehensive overview of potential genetic mechanisms that could explain the highly heterogeneous phenotype presented by the different carriers in the family. Given the wide audience of Nature Communications, I would recommend mentioning the following points in the discussion:

- exome sequencing does not inform on variants outside exons that can affect gene regulation (e.g., variants affecting promoters, enhancers, genome structure);
- from the methods provided, it is unclear if the authors investigated the potential contribution of copy number variation and/or structural variants, it would be good to clarify this;
- another plausible mechanism worth mentioning is a compound heterozygous scenario combining the GLI2 variants studied by the authors and a second, currently unknown variant also affecting GLI2, which would be only present in two very early onset diabetic siblings. The authors could mention that future genetic studies of the family using other technologies may further elucidate the genetic mechanism driving the observed phenotypical range.

2. A RNA-seq analysis using only 2 biological replicates is not standard. Readers of the manuscript should

be more readily informed of the details of the authors' experimental design by providing this information in the figure legend of Figure 3b, not just in the methods.

Inês Cebola

Response to the reviewers' comments

We thank the reviewers for finding the revised manuscript now suitable for publication at *Nature Communications*. Below is a point-by-point response to the remaining concerns of reviewer #3 with our responses shown in blue.

Reviewer #3 (Remarks to the Author):

There are still a couple of minor points, which I believe could be better addressed:

1. In terms of the listed limitations of the study, the authors fall short of providing a comprehensive overview of potential genetic mechanisms that could explain the highly heterogeneous phenotype presented by the different carriers in the family. Given the wide audience of *Nature Communications*, I would recommend mentioning the following points in the discussion:

- exome sequencing does not inform on variants outside exons that can affect gene regulation (e.g., variants affecting promoters, enhancers, genome structure);
- from the methods provided, it is unclear if the authors investigated the potential contribution of copy number variation and/or structural variants, it would be good to clarify this;
- another plausible mechanism worth mentioning is a compound heterozygous scenario combining the GLI2 variants studied by the authors and a second, currently unknown variant also affecting GLI2, which would be only present in two very early onset diabetic siblings. The authors could mention that future genetic studies of the family using other technologies may further elucidate the genetic mechanism driving the observed phenotypical range.

The Discussion Section has been expanded. We added all the points suggested by the reviewer about the potential genetic mechanism driving the observed phenotypical range within the family in the revised discussion (see page 11).

2. A RNA-seq analysis using only 2 biological replicates is not standard. Readers of the manuscript should be more readily informed of the details of the authors' experimental design by providing this information in the figure legend of Figure 3b, not just in the methods.

The information has been included in the figure legend associated to new Figure 4 (see page 27).